# The thymocyte-specific RNA-binding protein Arpp21 provides TCR repertoire diversity by binding to the 3′-UTR and promoting *Rag1* mRNA expression

Meng Xu[1,2,28], Taku Ito-Kureha[3,28], Hyun-Seo Kang[4,5], Aleksandar Chernev[6], Timsse Raj[3], Kai P. Hoefig[1], Christine Hohn[3], Florian Giesert[7], Yinhu Wang[8], Wenliang Pan[9], Natalia Ziętara[3,26], Tobias Straub[10], Regina Feederle[11], Carolin Daniel[12,13,14], Barbara Adler[15], Julian König[16], Stefan Feske[8], George C. Tsokos[9], Wolfgang Wurst[7,17,18,19], Henning Urlaub[6,20,21,22], Michael Sattler[4,5], Jan Kisielow[23,27]✉, F. Gregory Wulczyn[24]✉, Marcin Łyszkiewicz[1,25,29]✉ & Vigo Heissmeyer[1,3,29]✉

The regulation of thymocyte development by RNA-binding proteins (RBPs) is largely unexplored. We identify 642 RBPs in the thymus and focus on Arpp21, which shows selective and dynamic expression in early thymocytes. Arpp21 is downregulated in response to T cell receptor (TCR) and $Ca^{2+}$ signals. Down-regulation requires Stim1/Stim2 and CaMK4 expression and involves Arpp21 protein phosphorylation, polyubiquitination and proteasomal degradation. Arpp21 directly binds RNA through its R3H domain, with a preference for uridine-rich motifs, promoting the expression of target mRNAs. Analysis of the Arpp21−bound transcriptome reveals strong interactions with the *Rag1* 3′-UTR. Arpp21−deficient thymocytes show reduced Rag1 expression, delayed TCR rearrangement and a less diverse TCR repertoire. This phenotype is recapitulated in *Rag1* 3′-UTR mutant mice harboring a deletion of the Arpp21 response region. These findings show how thymocyte-specific Arpp21 promotes Rag1 expression to enable TCR repertoire diversity until signals from the TCR terminate Arpp21 and Rag1 activities.

RNA-binding proteins (RBPs) are essential for gene regulation in immune cells, and loss-of-function mutations of several RBPs are associated with autoimmunity and autoinflammation[1–7]. Although there is abundant information on the epigenetic and transcriptional gene regulation during thymocyte development, very little is understood about RBP-mediated post-transcriptional regulation, despite its well-characterized role in mature T cells[8].

T lymphocytes recognize antigens through the T cell receptor (TCR). A highly diverse TCR repertoire allows to clonally recognize a vast number of antigens and protect from pathogens. The generation of TCRs occurs in the thymus and involves the assembly of V(D)J elements in the TCRβ and α loci. The somatic rearrangement of V(D)J gene segments is mediated by a hetero-tetrameric complex of the recombination-activating genes Rag1 and Rag2[9]. As TCR recombination leads to double-strand breaks in DNA, the expression of *Rag* genes must be tightly controlled and restricted to specific stages to ensure the separation of recombination and proliferation in thymocytes and avoid genomic instability[10]. In conventional αβ T cells, expression of

A full list of affiliations appears at the end of the paper. ✉e-mail: jk@repertoire.com; gregory.wulczyn@charite.de; marcin.lyszkiewicz@uni-ulm.de; vigo.heissmeyer@med.uni-muenchen.de

*Rag* genes is restricted to two stages[11]. The first wave of Rag expression occurs in DN3 thymocytes when the TCRβ chain is assembled. After β-selection, Rag expression is transiently suppressed to block further recombination during expansion of the selected clones. Later, *Rag* gene expression is re-induced at the DP stage to recombine the TCRα chain and form a complete TCR. After successful recombination, Rag expression is completely suppressed and never restored in mature T cells[12]. Mutations that inactivate RAG1 or RAG2 function in humans or knockout mice prevent antigen receptor rearrangement and block T and B lymphocyte development, leading to severe combined immunodeficiency (SCID). Mutations causing hypomorphic RAG1 or RAG2 function have been associated with Omenn syndrome (OS) or combined immunodeficiency (CID)[13]. These primary immunodeficiencies (PIDs) show fewer defects in lymphocyte development, but more severe immune dysregulation[13]. Partial RAG activity leads to less efficient recombination and fewer lymphocyte clones are released into the periphery where they can undergo oligoclonal expansion. Reduced TCR diversity and, in particular, reduced usage of distal TCR alpha gene segments have been observed. In patients with hypomorphic RAG function or their corresponding mouse models, T and B cells can induce autoimmunity or severe inflammation[13].

Arpp21 is the best-studied member of a family of RBPs that includes R3hdm1, R3hdm2, Arpp21 (R3hdm3), and R3hdm4, all of which have a conserved amino-terminal R3H-domain coupled to longer carboxy-terminal regions harboring low complexity and intrinsically disordered sequences. The Arpp21 protein is induced during brain development in mouse embryos[14], where it is expressed in a highly abundant short splice-isoform encoding for a protein of ~21 kDa[15] as well as in a less abundant long isoform of 100 kDa. In this communication, we will refer to the 21 kDa isoform as Arpp21^short and to the 100 kDa isoform as Arpp21, which is also known as the thymocyte-expressed Tarpp protein[16]. Arpp21/Tarpp expression has been detected in cells of the thymic cortex and during early steps of thymocyte development starting at the CD4^-CD8^- double-negative (DN)2 and ending at the CD4^+CD8^+ double-positive (DP) stage[16]. Nevertheless, the function of Arpp21 in thymocyte development has not been demonstrated. Mechanistic analyses suggested an interaction with RNA, but only for the long isoform of Arpp21 that contains a composite R3H/SUZ RNA-binding domain[14]. Arpp21 was also shown to bind components of the EIF4F complex and tethering of Arpp21 to the 3′-UTR increased reporter expression, presumably by enhancing translation[14].

Here, we show that Arpp21 facilitates Rag1 expression at the post-transcriptional level, thereby enhancing TCR repertoire diversity in thymocytes. The R3H domain of Arpp21 preferentially binds oligo-U elements in vitro and Arpp21 interacts with such elements in the 3′-UTR of *Rag1* in vivo. Arpp21 and the Arpp21-responsive region in the 3′-UTR of Rag1 are similarly required for full Rag1 expression, efficient TCR rearrangement and the formation of a highly diverse TCR repertoire. Store-operated $Ca^{2+}$ entry (SOCE) through $Ca^{2+}$ release-activated $Ca^{2+}$ (CRAC) channels activates CaMK4 pathway-mediated Arpp21 phosphorylation, polyubiquitination and proteasomal degradation. Our findings describe a regulatory circuit in which Arpp21 promotes Rag1 expression to increase recombination efficiency and TCR repertoire complexity, with a negative feedback loop eliminating Arpp21 upon the completion of recombination to enable TCR-induced progenitor expansion in the absence of Rag1.

## Results

### Arpp21 is part of the thymic RBPome

We aimed to identify RBPs with specific functions in thymocytes. Using orthogonal organic phase separation (OOPS) we determined RNA-bound proteins in mouse thymocytes after UV-crosslinking and phenol/chloroform extraction. Protein/RNA adducts separate to the interphase between organic and aqueous phases[17] and after several

rounds of enrichment these proteins were released into the organic phase by RNase digestion (Fig. 1a). Mass spectrometry identified 642 proteins in total (Fig. 1b), among which 479 proteins (~75%) were already listed in the mouse eucaryotic RBP data base EuRBPDB[18]. To detect cross-linked RNA oligonucleotide moieties on RNA-binding proteins we used high-resolution mass spectrometry and automated analysis of the resulting mass spectra[19] (Supplementary Fig. 1). We confirmed cross-linked peptides for known RBPs in or close to the RNA-binding domain (RBD) (Supplementary Data 1) and used this analysis to confirm the direct interaction of 156 proteins with RNA (red dots in Fig. 1b) in addition to the confirmation of being present in the EuRBPDB (yellow dots in Fig. 1b). Comparing this OOPS dataset with a corresponding dataset obtained with in vitro expanded CD4^+ T cells[20], we noticed less thymic RBPs, but a comparable 75–80% overlap with the EuRBPDB (Fig. 1c). We focused on thymocyte-identified RBPs that were not detected in peripheral T cells (195 proteins). Among these were several DEAD-box helicases (Ddx21, Ddx39b, Ddx3x, Ddx46), several RNA-binding motif-type RBPs (Rbm25, Rbm34, Rbm39, Rbmx, Rbmx2) and many serine/arginine-rich splice factors (SRSF) that also harbor RRM domains (Srsf10, Srsf2, Srsf3, Srsf5, Srsf6, Srsf7). We also found Arpp21, an R3H/SUZ domain containing member of the R3hdm RBP family. Inspecting the ImmGen database (http://rstats.immgen.org/Skyline/skyline.html) for expression of these candidates in thymocytes[21], we found upregulation of mRNAs encoding for specific RBPs (Rbmx, Rbmx2), splicing factors (Srsf3, U2af1) as well as RNA methyltransferase (Nsun2), and a strong induction as well as selective expression only in thymocytes for Arpp21. Arpp21 was recently determined to be an RNA-binding protein regulating dendrite branching in neurons[14], and we also detected RNA moieties crosslinked to Arpp21 peptides in thymocytes (Supplementary Data 1). Arpp21 was also found upregulated during early thymocytes development[16] in a cluster of genes associated with T cell lineage commitment and TCR rearrangements that also included Tcf7/Lef1 and Rag1[21]. Arpp21 has thus been suggested to mediate key transitions in thymocytes differentiation[21]. These findings prompted us to explore Arpp21-mediated gene regulation in thymocytes. We confirmed selective protein expression in extracts from mouse brains and thymi (Fig. 1d). While two Arpp21 isoforms are present in the brain (Fig. 1e), with a predominance of Arpp21^short, the strong expression of Arpp21 in the thymic tissue was selective for the long isoform (Fig. 1d, e). Analyzing *Arpp21* mRNA, we found a steep increase from DN1 to DN2/3, a steep decline at DN4 as well as a second peak of high expression in DPs. Afterwards, *Arpp21* mRNA was almost undetectable in SP4 and SP8 cells (Fig. 1f). Although high *Arpp21* expression preceded the induction of *Rag* expression at DN2, this bimodal expression was very much similar to the known induction of *Rag1* and *Rag2* mRNAs at DN3 when gene segments in the TCRβ chain are rearranged and again at the DP stage when the TCRα chain is recombined (Fig. 1f)[10].

### TCR and $Ca^{2+}$ signaling induce degradation of Arpp21

Having shown that *Arpp21* mRNA starts to be expressed in DN2/DN3 and disappears after the DP stage (Fig. 1f), we asked whether TCR signaling affects Arpp21 protein expression. Using thymocytes from OTI transgenic mice in stimulations with the cognate $OVA_{257-264}$ SIINFEKL peptide, we found decreased Arpp21 protein expression as soon as after 1 h of stimulation (Fig. 2a). Even stronger decreases were induced by stimulation with anti-CD3 and anti-CD28 (Fig. 2b). TCR signaling activates PLCγ1 to cleave the $PIP_2$ substrate into $IP_3$ and DAG products, which trigger $Ca^{2+}$ and PKC signaling[22]. We mimicked $IP_3$ or DAG signaling pharmacologically by treating thymocytes with either ionomycin (iono) or phorbol ester (PMA), respectively. Ionomycin-treated thymocytes showed strongly decreased Arpp21 protein levels, which were similarly reduced after PMA/ionomycin but not PMA stimulation (Fig. 2c). We next asked whether $Ca^{2+}$-signaling induced degradation of Arpp21 by the proteasome. Indeed, pretreatment of

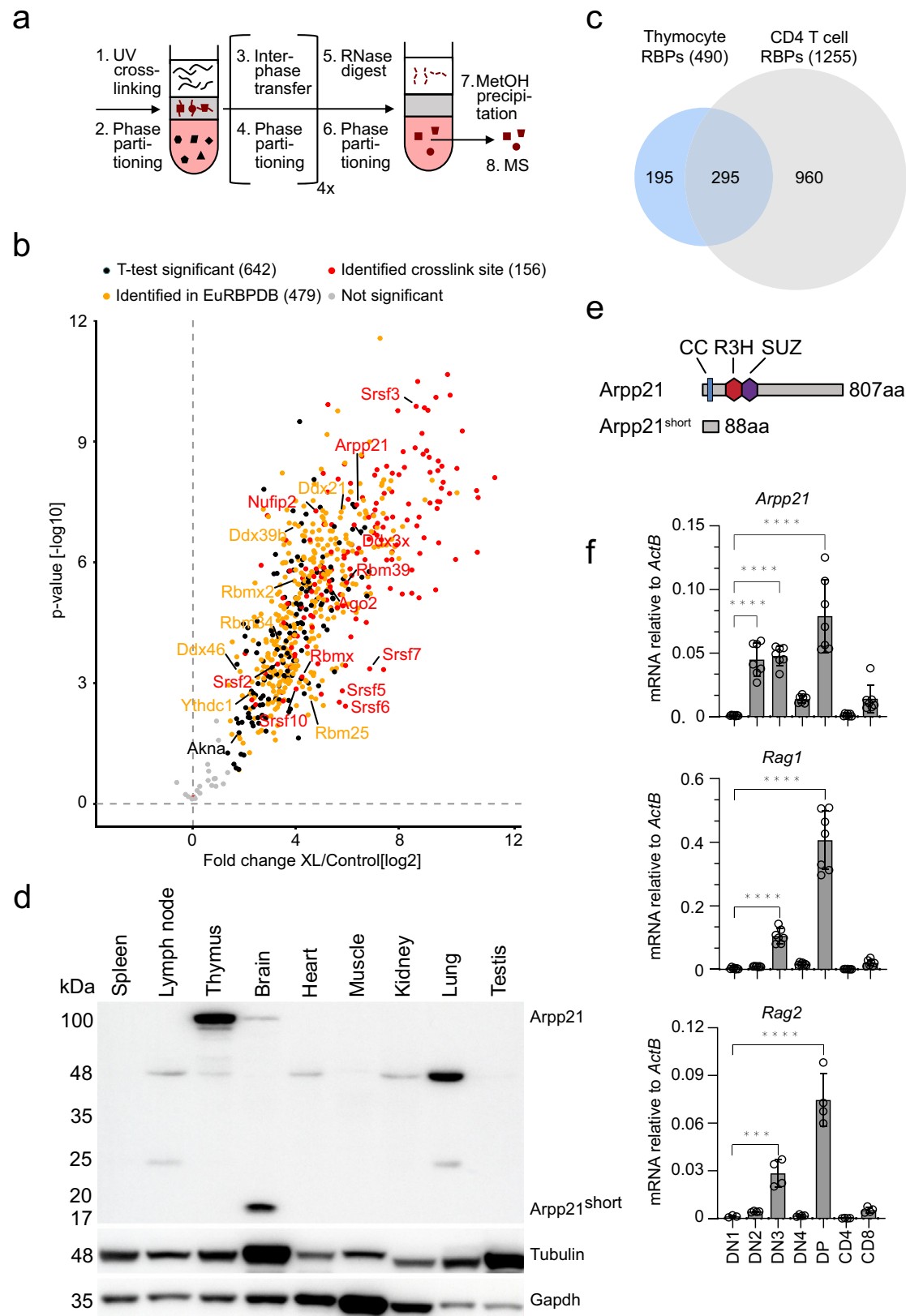

thymocytes with the proteasome inhibitor MG-132 prevented the $Ca^{2+}$-induced degradation of Arpp21 and resulted in accumulation of the Arpp21 band at a higher molecular weight (Fig. 2d). We then tested phosphorylation of the Arpp21 protein in anti-Arpp21 immunoprecipitates from extracts of MG-132/iono-treated thymocytes. Phosphatase treatment of these precipitates resulted in increased mobility of

Arpp21 rendering its migration indistinguishable from untreated controls (Fig. 2e). Furthermore, probing immunoblots of anti-Arpp21 immunoprecipitations with ubiquitin-specific antibodies revealed polyubiquitination of the Arpp21 protein that was stabilized due to proteasome inhibition in MG-132-pretreated and ionomycin-stimulated cells (Fig. 2f). We further dissected if $Ca^{2+}$ release from

**Fig. 1 | The thymic RBPome includes the dynamically expressed Arpp21 protein. a** Schematic representation of the OOPS method. **b** Volcano plots from two-sided Student's *t* test analysis using a permutation-based FDR method for multiple hypothesis corrections showing the −log 10 *p* value plotted against the log2 fold-change comparing the organic phase after RNase treatment of the interphase of OOPS experiments of the crosslinked (XL) mouse thymocytes versus the non-crosslinked sample. Black dots represent proteins significant at a 5% FDR cutoff level. Yellow dots show RBPs additionally identified in the EuRBPDB and red dots represent proteins for which crosslinks to RNA moieties have directly been measured. **c** Venn diagram showing the overlap between mouse CD4 T cell RBPs and

confirmed RBPs detected in thymocytes (represented by yellow and red dots in (**b**)). **d** Representative immunoblots from two independent experiments showing the indicated proteins in cell lysates from different tissues. **e** Scheme of Arpp21 isoforms showing the domains of R3H in red, SUZ in purple and coiled-coil (CC) in blue. **f** RT-qPCR analysis of the indicated mRNAs in sorted DN1, DN2, DN3, DN4, DP, CD4-SP, or CD8-SP thymocytes from WT mice. Data are presented as mean values ± SD. *n* = 7 (for Arpp21 and Rag1) or *n* = 4 (for Rag2), statistic significance was determined by One-way ANOVA, *$p < 0.05$, **$P < 0.01$, ***$P < 0.001$ and ****$P < 0.001$. Source data for (**d**, **f**) are provided as a Source Data file.

the ER or CRAC channel-dependent $Ca^{2+}$-influx across the plasma membrane are involved in the regulation of Arpp21 levels. The loss of Arpp21 in thymocytes triggered by ionomycin stimulation was only observed in conditions when extracellular $Ca^{2+}$ was present, but not in cells cultured in $Ca^{2+}$-free Ringer solution (Supplementary Fig. 2a) indicating that Arpp21 levels are regulated by CRAC channels and SOCE. We then tested the CRAC channel-specific inhibitor BTP2[23], which at concentrations as low as 0.3 μM caused stabilization of Arpp21 in its non-phosphorylated form in ionomycin-stimulated cells (Supplementary Fig. 2b) Underscoring the dependency on SOCE, we tested Arpp21 degradation in thymocytes lacking expression of STIM1 and STIM2, two ER-resident $Ca^{2+}$ sensors involved in CRAC channel activation. Thymocytes from *Stim1/Stim2* DKO mice did not support Arpp21 degradation in response to ionomycin stimulation (Fig. 2g) and neither DN nor DP thymocytes from these mice showed SOCE (Supplementary Fig. 2c, d). Testing inhibitors of $Ca^{2+}$-dependent protein kinases we found that 10 μM KN-93 prevented Arpp21 degradation, suggesting CaMK4 involvement (Supplementary Fig. 2e)[24]. Importantly, thymocytes from CaMK4-deficient mice did not show either retarded migration or degradation of Arpp21 after stimulation with ionomycin (Fig. 2h). Together, these data reveal that Arpp21 expression is suppressed by SOCE and CaMK4 phosphorylation-induced proteasomal decay.

## Interaction of Arpp21 with the transcriptome of thymocytes

To investigate the function of Arpp21 in thymocytes, we generated mice carrying a targeted deletion in the *Arpp21* locus. To minimize effects on the downstream intron-embedded miR-128-2 gene, we designed two sgRNAs to excise exon 5 and cause a frame-shift in the open reading frame of the long isoform of *Arpp21* (Fig. 3a). We established a mouse line in which such a deletion resulted in the complete loss of full-length Arpp21 protein (Fig. 3b), leaving miR-128 expression intact in thymocytes as well as thymocyte subsets (Supplementary Fig. 3a, b). Arpp21-deficient mice were healthy and fertile and showed no apparent abnormalities within the first year of life. Targeting of the *Arpp21* locus did not give rise to non-naturally occurring protein variants. Instead, and presumably due to increased alternative splicing or promoter usage we found increased Arpp21^short protein levels in extracts from Arpp21-deficient thymocytes (Fig. 3b).

To identify mRNA targets we UV-irradiated thymocytes and covalently crosslinked Arpp21 to RNA. After protein extraction and partial RNase digestion we performed immunoprecipitations with Arpp21-specific antibodies followed by sequencing of Arpp21-linked RNA fragments (Fig. 3c and Supplementary Fig. 3c). As a control we performed the same immunoprecipitation with extracts from Arpp21 knockout thymocytes. To additionally estimate the background, we also prepared libraries of RNA fragments in the identical size range of non-immunoprecipitated wild-type samples (size-matched input, SM-Input). The radioactive smear of Arpp21/RNA-adducts was strongly reduced in the knockout but appeared upwards from 100 kDa in the wild-type sample, likely representing the long isoform of Arpp21 protein crosslinked to RNA fragments (Fig. 3c). Sequencing reads exceeded 30 million per sample in all four replicates and identified 6779 direct Arpp21 target mRNAs in thymocytes. The peaks displayed a

strong enrichment for the 3′-UTRs of mRNAs (Fig. 3d). Bound mRNAs displayed a moderate enrichment for miR-128-3p binding sites, as previously suggested[14], but also for the haematopoietic and thymus-expressed miRNA miR-181a[25,26], indicating that Arpp21 per se may preferentially recognize mRNAs subject to miRNA-dependent regulation (Supplementary Fig. 3d). We then determined changes in the transcriptome in flow cytometry-enriched DN2 and DN3 thymocytes from WT and Arpp21-deficient mice. We identified 67 differentially expressed genes (DEGs, padj < 0.05), of which the majority (56 DEGs) were downregulated with only few (11 DEGs) upregulated mRNAs at the DN3 stage (Fig. 3e). Of note, the DEGs were less prominent at the DN2 stage (Supplementary Fig. 3e). Comparing the DEGs with the iCLIP results we determined an intersection of 21 downregulated genes (Fig. 3e, f, g) and two upregulated mRNAs, which, however, exhibited much lower Arpp21 binding (Supplementary Fig. 3f). The mRNAs bound by the endogenous Arpp21 protein in thymocytes overlapped with those bound by overexpressed tagged Arpp21 in HEK293 cells[14] including *Phf6, Lin7c* and TGFβ pathway genes such as *Tgfbr1* (Fig. 3f). For two targets of Arpp21, the *Rc3h1* and *Rc3h2* mRNAs, that encode the Roquin-1 and Roquin-2 RBPs with known immune functions, we analyzed if endogenous Roquin proteins showed regulation in response to Arpp21 expression. Indeed, only Arpp21 but not Arpp21^short promoted endogenous Roquin-1 or Roquin-2 protein expression as shown by intracellular anti-Roquin-1/2 staining in GFP^+-gated MEF cells that expressed GFP-Arpp21 (Supplementary Fig. 3g). We also confirmed upregulation of the respective endogenous Roquin-1 or Roquin-2 proteins by immunoblots in different MEF cell lines upon ectopic Arpp21 expression (Supplementary Fig. 3h). Most interestingly, we identified the thymocyte-expressed *Rag1* mRNA as an Arpp21 target, with a read coverage of crosslinking events per peak that was almost 100-fold higher compared to the other Arpp21 targets (Fig. 3f). Considering only iCLIP targets, *Rag1* was also among the most down-regulated mRNAs in Arpp21-deficient DN3 cells (Fig. 3g). These data prompted us to investigate post-transcriptional regulation of *Rag1* by Arpp21 in thymocyte development.

## The R3H domain of Arpp21 binds uridine-rich RNA elements

We identified U-rich sequences as top Arpp21-bound motif (Fig. 4a). Using peak calling and PureCLIP determination of binding sites at near-nucleotide resolution[27] we discriminated at least 8 binding sites for Arpp21 in the 3′-UTR of *Rag1* characterized by high read density and reoccurring truncations (Fig. 4b). Such truncations correlate to protease-resistant peptide adducts at crosslink sites in the RNA causing premature termination during reverse transcription. Similar to other Arpp21 targets[14], peaks in the 3421nts long 3′-UTR of *Rag1* mapped to extended U-rich sequences (Fig. 4c).

We set out to define structural determinants of Arpp21 binding to RNA. To do so we generated various N-terminal fragments of Arpp21, all containing the predicted R3H region, while the full SUZ domain (residues 229-298) was only present in Arpp21^1-330 (Fig. 4d). We analyzed Arpp21^61-260 and Arpp21^130-260 by $^1H/^{15}N$-correlation NMR spectrum (HSQC), since these constructs as compared to longer protein fragments showed well-dispersed signals corresponding to structured regions (Supplementary Fig. 4a), Arpp21^61–260 had similar binding

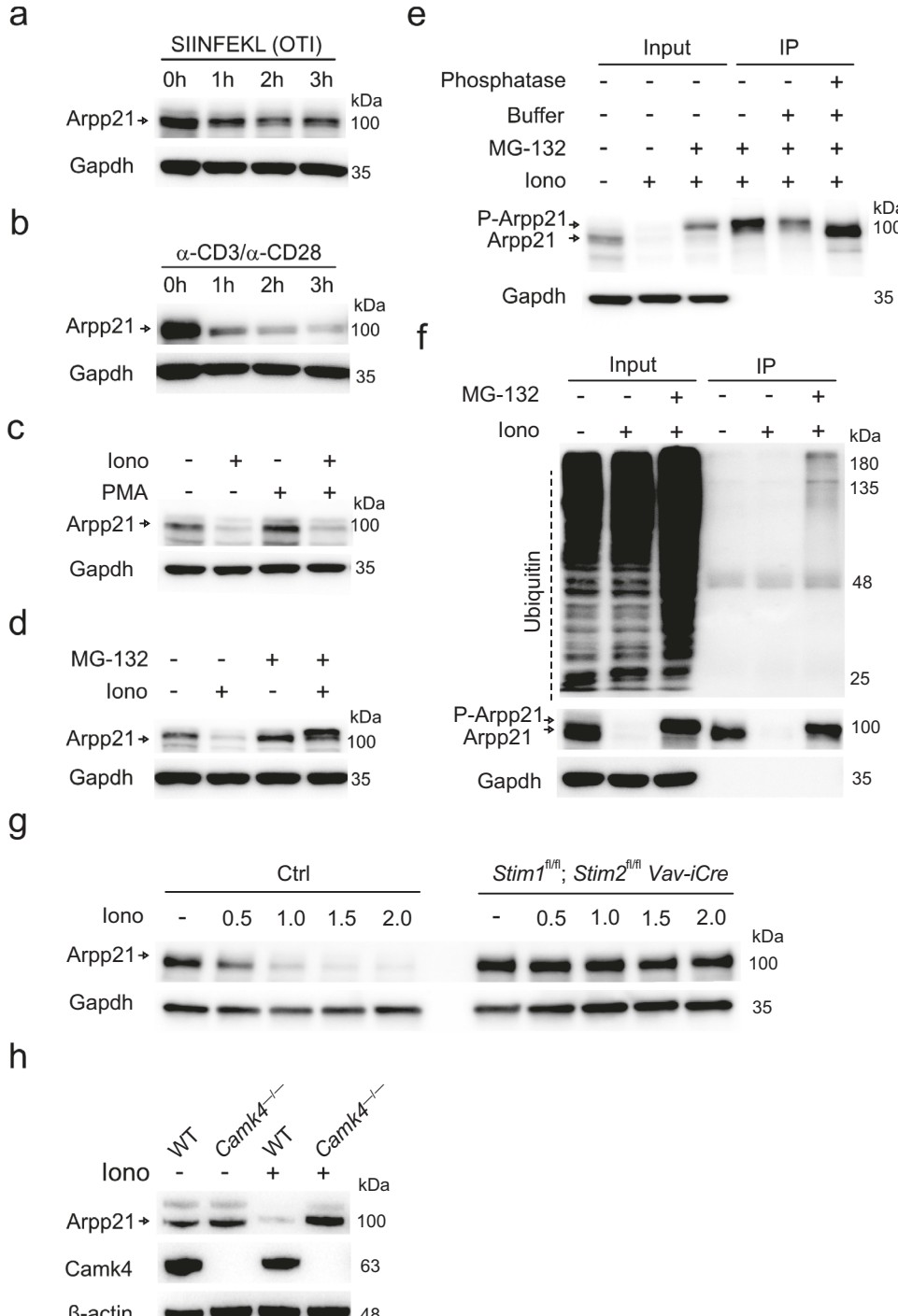

**Fig. 2 | Ca²⁺ signals induce phosphorylation, polyubiquitination and proteasomal degradation of Arpp21.** Immunoblot analyses of Arpp21 and indicated loading controls in extracts of thymocytes from (**a**) TCR-transgenic OTI, *Stim1*^fl/fl^;*Stim2*^fl/fl^;*Vav-iCre* (**g**) or *Camk4*^−/−^ (**h**) as well as WT mice (**b**–**h**). Cells were stimulated with (**a**) 2 μg/mL of cognate peptide SIINFEKL or (**b**) 1 μg/mL of anti-CD3 and 2 μg/mL anti-CD28 for 0 h, 1 h, 2 h and 3 h or (**c**–**h**) with 1 μM ionomycin or (**c**) 20 nM PMA or (**d**–**f**) 10 μM MG132, as indicated. **e**, **f** Protein extracts from WT mice

stimulated with ionomycin and MG-132 were immunoprecipitated with Arpp21-specific antibodies, separated by SDS-PAGE and probed with ubiquitin-reactive antibodies or treated with phosphatase buffer and with antarctic phosphatase to analyze Arpp21 mobility in SDS-PAGE as detected by immunoblotting. Each experiment was performed at least twice. Source data for (**a**–**h**) are provided as a Source Data file.

affinity as compared to the larger N-terminal construct Arpp21^1-330^ for U9 RNA oligo in isothermal titration calorimetry (ITC) (Supplementary Fig. 4b). NMR-monitored RNA titrations of Arpp21^61-260^ and Arpp21^130-260^ showed identical spectral changes upon addition of oligo-U RNA (Supplementary Fig. 4c, d). The significant spectral changes upon adding U9 RNA, both in resonance shifting (CSP) and broadening

(intensity loss), were observed mainly for the dispersed signals, corresponding to the structured region (Fig. 4e). At the equimolar point, many resonances appeared at defined new positions, suggesting a stable bound conformation. Using triple resonance NMR experiments, we assigned ~50% resonances in the minimal Arpp21^130-260^ protein including N-/C-terminal regions and the R3H domain (Fig. 4e), defining

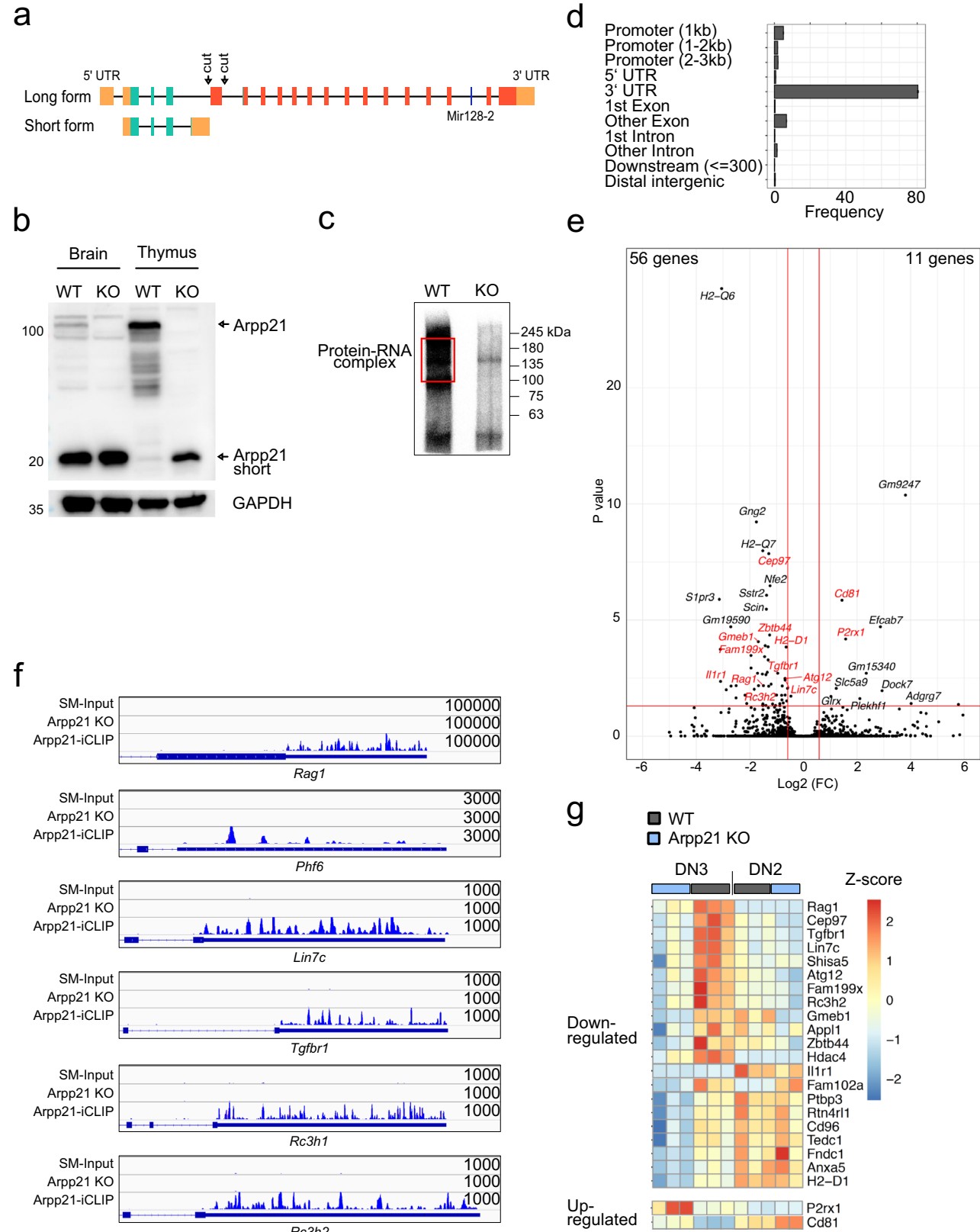

the key region and residues involved in RNA-binding. In the absence of RNA, we observed overall rigid backbone dynamics for Arpp21$^{130-260}$, except for its N- and C-termini (Supplementary Fig. 4c). Finally, we performed NMR titrations to address the binding specificity of Arpp21$^{130-260}$ for U6, A6, C6 or G6 RNA oligos. Consistent with the iCLIP

results, Arpp21$^{130-260}$ spectral changes were only observed in the presence of U6 RNA (Fig. 4e, Supplementary Fig. 5a, b), but not for A6/C6/G6 RNAs (Fig. 4f, Supplementary Fig. 5c, e). Together, these findings demonstrate the ability of the R3H domain of Arpp21 to bind RNA and reveal a domain-intrinsic specificity for uridine-rich sequences.

**Fig. 3 | Identification of targets and binding sites of Arpp21 in thymocytes.**
**a** Description of the *Arpp21/miR-128-2* gene locus, indicating the *Arpp21* gene targeting strategy. **b** Representative immunoblot analysis of Arpp21 and Arpp21short proteins in brain and thymus extracts of WT and *Arpp21*−/− mice. The experiment was repeated twice. **c** Representative autoradiography of crosslinked and [32]P-labeled RNA-Arpp21 immunoprecipitation complexes from thymocyte lysates of WT and *Arpp21*−/− mice (*n* = 4 each). **d** Association of Arpp21 peaks on RNA transcript according to gene region. **e** Volcano plot of DEGs identified in mRNA-sequencing of DN3 thymocytes showing the −log10 (*P* value) plotted against the log2 (fold-change). The Arpp21-target mRNAs identified by iCLIP are shown in red. **f** Integrated genome view displaying Arpp21 iCLIP reads in the 3′-UTRs of *Phf6*, *Lin7c*, *Tgfbr1*, *Rc3h1*, *Rc3h2*, and *Rag1* showing merged data from four biological replicates. **g** Heat map showing differentially expressed Arpp21-target genes identified in mRNA-sequencing of DN3 thymocytes (padj < 0.1) in the intersection with iCLIP data in DN2 and DN3 thymocytes from WT (gray) and *Arpp21*−/− (blue) mice. Values represent the z-score. Source data for (**b**) are provided as a Source Data file.

## Arpp21-deficient thymocytes show a developmental block

We then analyzed T cell development in Arpp21−deficient mice. Here, we detected comparable frequencies of the main thymocyte populations (DN, DP, SP4 and SP8) to wild-type mice (Fig. 5a). However, loss of Arpp21 resulted in the accumulation of DN3 (lin−CD44−CD25+) at the expense of DN4 (lin−CD44−CD25−) cells, suggesting a partial developmental block (Fig. 5b) that corresponds with the developmental stages when Arpp21 is normally induced (Figs. 1e, 5c). We generated a new monoclonal antibody that specifically recognized only the long isoform of Arpp21 that binds to RNA (Supplementary Fig. 6). This antibody (8G2) was specific only for this paralog and this isoform as demonstrated by intracellular anti-Arpp21 staining in MEF cells expressing GFP-tagged versions of Arpp21, Arpp21short, R3hdm1, R3hdm2 or R3hdm4 proteins (Supplementary Fig. 6a, b). Importantly, with this antibody we confirmed the degradation of Arpp21 in response to ionomycin stimulation of thymocytes using immunoblots and flow cytometry (Supplementary Fig. 6c, d). We then analyzed the kinetics of Arpp21 protein expression during pre-TCR signal transduction and positive selection stages (Fig. 5c). Arpp21 was expressed at high levels from DN2 to DN4, with a notable decline between DN3c and DN4b. Positive selection in thymocytes (TCRβmedCD69+) was correlated with strongly reduced Arpp21 protein expression (Fig. 5c). This observation prompted us to do a detailed analysis of DN subsets by following the frequency of all major DN progenitors. We discovered a substantial accumulation of DN3a (lin−CD117−CD44−CD25hiCD28lo) progenitors, and a significant reduction of DN4a+b (lin−CD117−CD44−CD25−CD28hi) cells (Fig. 5d). The frequencies of the pluripotent early thymic progenitors (ETP, lin−CD117hiCD44hiCD25−) and DN2a cells (lin−CD117hiCD44hiCD25+) were not altered, suggesting that Arpp21 does not play a role in T cell lineage commitment. Of note, the accumulation of DN3a progenitors, which is a population on the verge of TCRβ-selection, suggests regulatory potential for Arpp21 in this process.

We also analyzed B cell development in the bone marrow (Supplementary Fig. 7a), where we found a smaller reduction in the total B cells and PreB2 (CD11c−B220+ CD25+CD117−) cells in Arpp21−deficient compared to WT mice (Supplementary Fig. 7b). *Arpp21* mRNA was expressed at relatively higher levels in PreB1 (CD11c−B220+CD117+CD25−) cells, where it coincided with high Rag1 mRNA levels in wild-type mice (Supplementary Fig. 7c). However, analyzing bone marrow cells and thymocytes by flow cytometry we found that Arpp21 protein expression was markedly lower in the bone marrow as compared to thymocytes (Supplementary Fig. 7d). Accordingly, Rag1 mRNA levels and intracellular IgM expression did not show significant differences at the early stages of B cell development in Arpp21−deficient mice (Supplementary Fig. 7e, f). Together these data suggest a more prominent role of Arpp21 in early T cell and a lesser and Rag1-independent importance in B cell development.

## Arpp21 promotes *Rag1* expression through its 3′-UTR

We also investigated the Arpp21-dependent regulation of Rag1 in the thymus. We showed that the mRNA of *Rag1*, but not *Rag2*, was significantly decreased in DN3 thymocytes from *Arpp21*−/− mice compared to WT counterparts (Fig. 6a). In thymocyte extracts Rag1 protein was also significantly reduced in *Arpp21*−/− compared to WT

mice (Fig. 6b). Reporter assays in HeLa cells were performed to determine the effect of Arpp21 overexpression on the *Rag1* 3′-UTR. Using a dual-luciferase system we found that the expression of the *Rag1* 3′-UTR reporter increased in response to Arpp21 overexpression (Fig. 6c, left panel). No impact was detected for the *Rag2* 3′-UTR or empty reporters (Fig. 6c, left panel). Using RT-qPCR we confirmed Arpp21−induced upregulation of the reporter on the mRNA level, however, we detected smaller effects (Fig. 6c, right panel), supporting stronger contributions on the protein level, as previously proposed[14]. Using progressive 3′ end deletion of the *Rag1* 3′-UTR we mapped relative contributions to Arpp21-mediated regulation by individual regions (Fig. 6d, upper panel). Deletion of 330 nts at the 3′ end of the transcript led to a significant, but incomplete drop in Arpp21-mediated regulation (Mut1). Likewise, further, progressive deletions (Mut2-4) strongly reduced but did not eliminate the Arpp21 response until removal of all but the initial 469 nts (Mut5) (Fig. 6d, lower panel). These findings showed that the 3′-UTR of *Rag1* is sufficient for Arpp21-mediated regulation, which is encoded in an extended region consistent with the distribution of iCLIP peaks along the entire 3′UTR (Figs. 3f, 4b).

## Arpp21-deficiency reduces repertoire diversity

Given the limited time thymic progenitors have to successfully recombine TCR beta and alpha loci to escape apoptosis, combined with the sequential and highly orchestrated order of recombination element selection, altered Rag1 protein expression should result in a skewed recombination process and consequently an altered TCR repertoire. We therefore hypothesized that associated with *Arpp21* deletion the impaired expression of Rag1 may impact on recombination of gene segments encoding the TCRβ chain at the DN3 stage[28]. Indeed, intracellular staining of DN3 cells revealed a reduced frequency of cells with TCRβ protein expression compared to DN3 thymocytes of WT mice (Fig. 6e). We then used semi-quantitative PCR to detect V(D)J rearrangement of the *Tcrb* gene. CD25+ DN (combining DN2 and DN3) thymocytes were sorted from WT and *Arpp21*−/− mice and different amounts of genomic DNA were used to determine Dβ2-to-Jβ2, Vβ8.2-to-DJβ2, Vβ11-to-DJβ2 and Vβ12-to-DJβ2 rearrangements[29]. As hypothesized, we detected reduced rearrangement in thymocytes from *Arpp21*−/− mice compared to WT mice, except for the early Dβ2-to-Jβ2 rearrangement (Fig. 6f). This finding suggested that Arpp21 deficiency reduces the rate of V(D)J recombination at the TCRβ locus. To investigate the significance of Arpp21 globally, we assessed the diversity of the TCR repertoire. To this end, Arpp21−sufficient and deficient flow cytometry-enriched splenic CD4+ T cells were subjected to TCRα and β next-generation sequencing (NGS). Consistent with the PCR results, the overall clonality of T cells from Arpp21−deficient mice was reduced to approximately 70% of WT counterparts (Fig. 7a). As a measure of repertoire diversity, we calculated the Shannon's diversity index, which takes into account both the number of total sequences detected and the clonal size distribution. Here, Arpp21-deficient T cells had a lower Shannon's index for TRBV but not for TRAV (Fig. 7b). To verify that the altered clonal output of the thymus results in clonal expansion, we assessed the cumulative percentage of TRBV clones. We found that the top 100 clones accounted for 3%

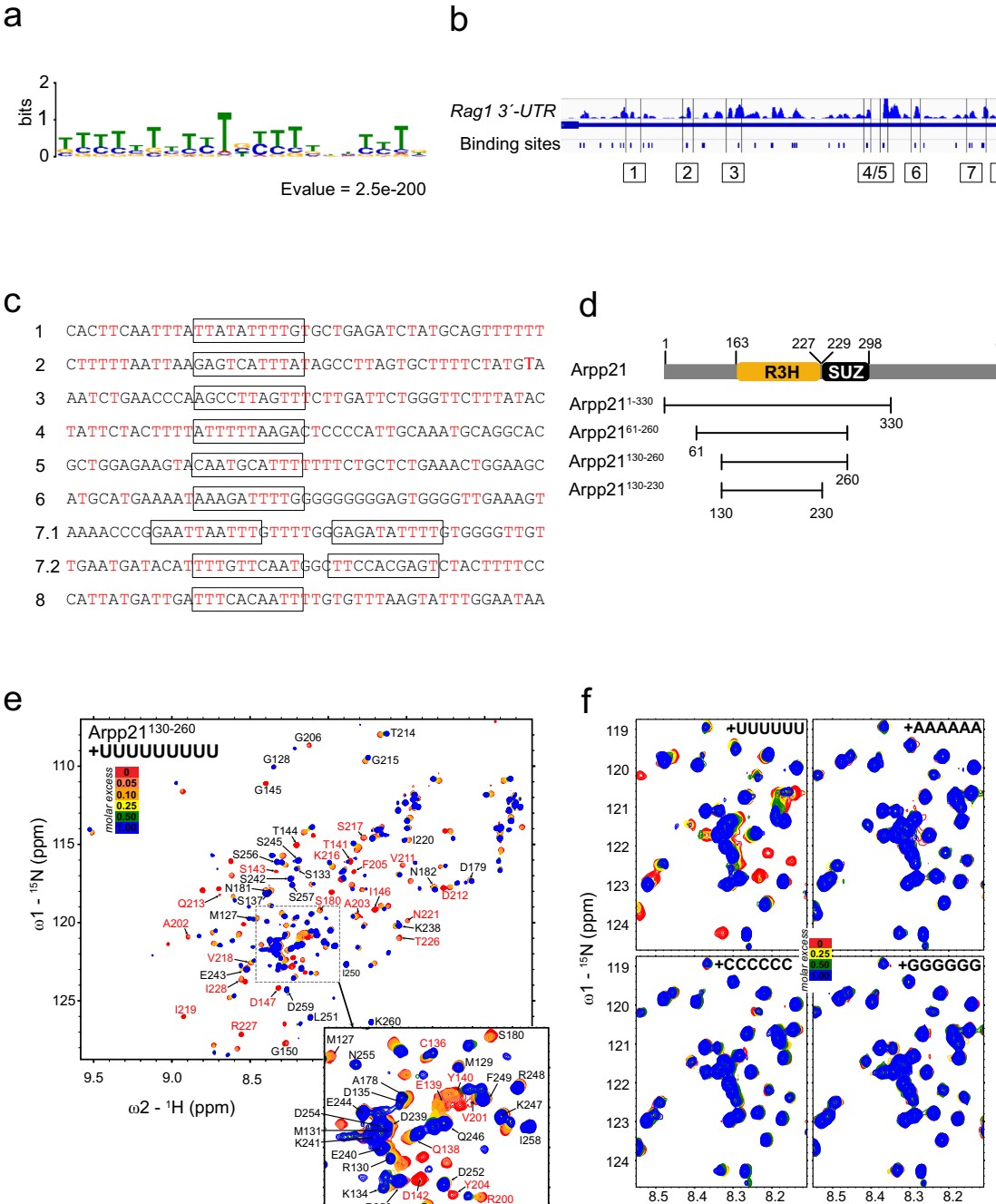

**Fig. 4 | Arpp21 binds uridine-rich motifs through its R3H domain. a** MEME motif search identifies a uridine-rich sequence enriched at Arpp21 binding sites. **b** Integrated genome view of Arpp21 iCLIP reads in the 3'-UTRs of *Rag1* with predicted binding sites called by PureCLIP. **c** Sequences in and flanking peak regions of the 3'-UTRs of *Rag1* are displayed. **d** Arpp21 subconstructs used for the NMR experiments within the N-terminal region in relation to R3H (163-226) or SUZ (227-298) domains as predicted by PROSITE. **e** HSQC-monitored titration of Arpp21[130-260] with U9 RNA (molar ratios of 0, 0.05, 0.1, 0.25, 0.5, 1.0). Assigned resonances are annotated and the resonances with significant CSP or intensity loss, at the equimolar U9 RNA addition, are annotated in red. **f** Comparisons of Arpp21[130-260] titrations with four types of RNA (U6, A6, C6, G6) with the protein-to-RNA molar ratios of 0, 0.25, 0.5, 1.0 (see Supplementary Fig. 5 for full spectra).

of the total number of TRBV sequences detected in WT mice, whereas this parameter increased to 6% in Arpp21–deficient mice (Fig. 7c). A similar pattern was observed when the top 20 TRBV clones were evaluated (Fig. 7d). By analyzing clonal diversity and the cumulative percentage of clones, it is possible to distinguish between events occurring during TCR recombination and peripheral clonal expansion. Knowing that overall recombination is impaired, we assessed whether Arpp21 deficiency affected individual V segments. Ordering productive TRBV and TRAV segments according to their position on the chromosomes, allowed to assess whether there was preferential usage of proximal variable elements (Fig. 7e–g). Of 22 TRBV segments, all but two were more frequently used in WT regardless of chromosomal location, suggesting strong alteration of TCR beta locus recombination in Arpp21–deficient mice (Fig. 7g). Although most of the 102 detected TRAV segments in the central and distal chromosomal fragments were preferentially used in WT mice, the most proximal segments were preferentially used in Arpp21–deficient mice (Fig. 7e, f). A similar phenomenon can be

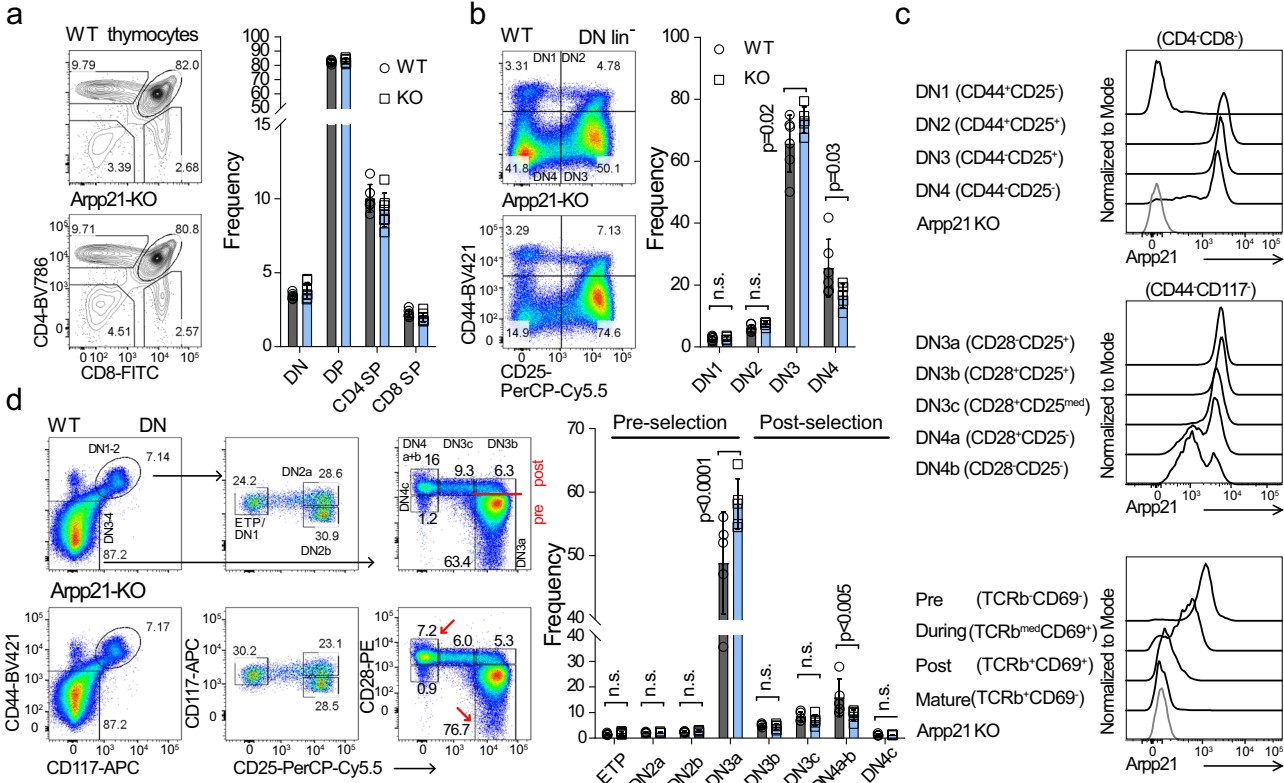

**Fig. 5 | Arpp21 deficiency results in the accumulation of DN3a thymic progenitors. a** Comparison of thymocyte development in WT and *Arpp21⁻/⁻* mice. Representative contour plots (left) and statistical analysis (right) of main thymic populations (WT *n* = 6; KO = 7). **b** Representative pseudo-color plots (left) and statistical analysis (right) of main DN populations (WT *n* = 6; KO = 7). **c** Flow cytometry of DN or DP thymocytes on the indicated subpopulations stained with 8G2 antibody. (*n* = 4). The stages of positive selection were defined by TCRβ and CD69 markers. **d** Pseudo-color plots showing all DN populations in WT and *Arpp21⁻/⁻* mice (left) and statistical analysis (right) (WT *n* = 5; KO = 5). Red arrows show either accumulation (DN3a) or reduced (DN4a+b) frequency of progenitors in the Arpp21-deficient thymus. Data are representative of three (**a**) or two (**b, d**) experiments. Each point represents an individual thymus, mean values with SD are shown. Statistical analysis was performed using two-way ANOVA followed by Sidak's multiple comparison test. Source data for (**a–d**) are provided as a Source Data file.

observed for the TRAJ segments, where the segments most proximal to the TEA are prominently used in Arpp21-deficient thymocytes at the expense of distal fragments (Fig. 7e, h). Considering that the TRAJ elements from 58 to 45 are preferentially used during the first round of recombination and that the use of the more distal elements is almost exclusively restricted to the subsequent rounds of recombination, it is evident that altered expression of Rag1 in the Arpp21 knockout has far-reaching consequences for the TCR repertoire. Finally, by examining the length of CDR3 elements, a process that is independent of Rag1, we found no changes regardless of Arpp21 status (Fig. 7i)[30–32].

In conclusion, deficiency of Arpp21 leads to reduced clonality and altered clonal output during T cell generation in the thymus. In addition, Arpp21 deficiency causes the preferential use of proximal fragments in TRAV and TRAJ segments significantly limiting the TCRA-elements choice.

### Arpp21-insensitive *Rag1* phenocopies Arpp21 deficiency

To remove binding sites and render *Rag1* insensitive to Arpp21 regulation we introduced a large, internal deletion in the *Rag1* 3′-UTR in the mouse germline. Crossing of two founders generated homozygous F1 offspring, which we termed *Rag1*³'del/3'del mice (Fig. 8a). Remarkably, thymocyte development showed impaired DN3-DN3b transition in *Rag1*³'del/3'del similar to *Arpp21⁻/⁻* mice (Fig. 8b, c), while the gross T cell development remained intact in both lines (Supplementary Fig. 8a and Fig. 5a). Consistent with Arpp21-deficient mice,

intracellular TCRβ expression in DN3 thymocytes was reduced (Fig. 8d and Fig. 6e). Similar to Arpp21-deficient thymocytes, we determined reduced Rag1 expression in immunoblots and RT-qPCR analyses, with stronger effects on the protein than mRNA levels (Fig. 8e, f and Fig. 6a, b). Having established that the *Rag1* 3′-UTR deletion recapitulates the thymic phenotype of *Arpp21⁻/⁻* mice, we also investigated the repertoire of peripheral T helper cells by TCR sequencing. The overall effects were comparable to *Arpp21⁻/⁻* mice albeit a bit smaller. No gross changes were observed in the number of clones detected or in the diversity index of either TCRα or TCRβ repertoires (Supplementary Fig. 8d, e), however, expansion of TRBV clones recapitulated the observation from Arpp21-deficient mice (Fig. 8g and Fig. 7d). Importantly, the use of variable segments showed similar changes as seen in *Arpp21⁻/⁻* mice. Again, similar to *Arpp21⁻/⁻* mice, the use of TRBV segments was generally altered in *Rag1*³'del/3'del T cells (Fig. 8h and Fig. 7e), regardless of their chromosomal location, while the more numerous TRAV and TRAJ fragments, showed preferential use of proximal fragments in the *Rag1*³'del/3'del cells (Fig. 8h, i, Supplementary Fig. 8f, g and Fig. 7e, f). Finally, and again consistent with the Arpp21 KO mice, the length of the CDR3 elements was unaltered (Supplementary Fig. 8h and Fig. 7i). These extended similarities in thymocyte development, TCR recombination and preferential usage of proximal TRAV and TRAJ segments in *Rag1*³'del/3'del and *Arpp21⁻/⁻* mice strongly suggest that the Rag1 3′-UTR contains regulatory sequences recognized by Arpp21 that are required for the full gamut of Rag-1 recombination activity.

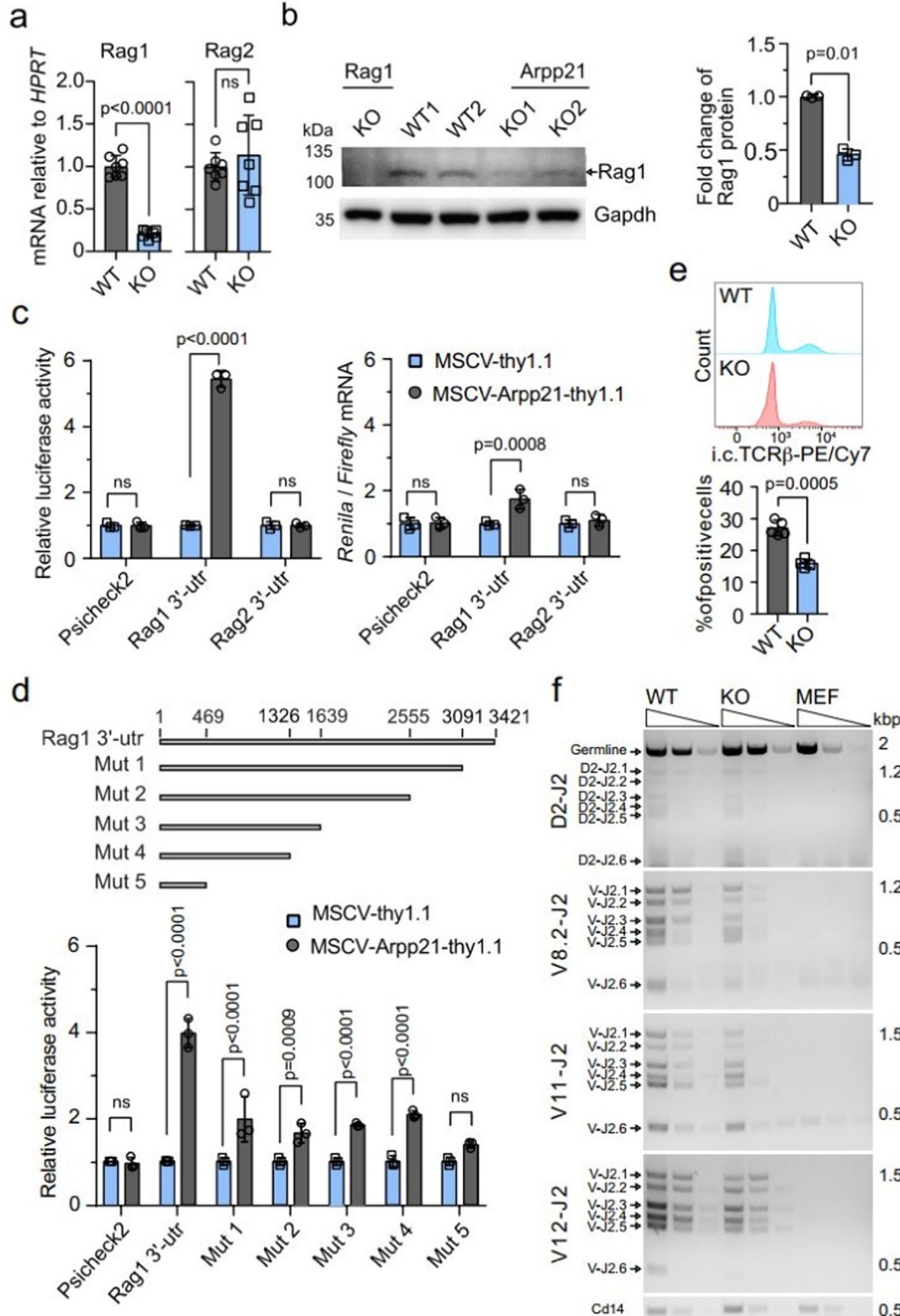

## Rag1 heterozygosity exposes post-transcriptional regulation of Rag1

To further test the relevance of Rag1 protein amounts for T cell development and the role of post-transcriptional regulation in its expression, we allowed Rag1 to be expressed from only one functional locus. To this end, we crossed either WT or $Rag1^{3del/3del}$ mice with Rag1-deficient mice to generate $Rag1^{+/-}$ and $Rag1^{3del/-}$ strains. We then tested Rag1 expression in thymocytes from these mice compared to WT, Arpp21-deficient and $Rag1^{3del/3del}$ mice in Western blots (Fig. 9a). The results suggest a clear dose-response relationship between Rag1 levels

**Fig. 6 | Arpp21-dependent regulation of the *Rag1* 3′-UTR promotes TCRβ rearrangement. a** RT-qPCR determining Rag1 and Rag2 mRNA levels in DN3 thymocytes from WT and *Arpp21⁻/⁻* mice. Data are presented as mean values ± SD. *n* = 7, statistic significance was determined by two-sided Student's *t* test, with ****P* < 0.001 indicating significance, and n.s. indicating no significance (*p* > 0.05). **b** Western blot analysis of Rag1 protein expression in whole thymocytes lysates from *Rag1⁻/⁻*, *WT* and *Arpp21⁻/⁻* mice (left) with a densitometric quantification of Rag1 protein expression in *WT* and *Arpp21⁻/⁻* thymocytes shown in (right), statistical significance was determined by two-sided Student's *t* test. **c, d** Hela cells were transduced with MSCV-thy1.1 or MSCV-Arpp21-thy1.1 retroviruses. Cells were then transfected with psiCheck2 empty (negative control) or psiCheck2 vectors harboring the indicated 3′-UTRs in full-length (**c**) or 3′end deleted versions (**d**). Renilla

and Firefly luciferase activity (upper panel) or mRNA level (lower panel) were assessed. Data are representative of two or more experiments. Data are represented as mean values ± SD. Two-way ANOVA was used to determine statistical significance and ns indicates no significance (*p* > 0.05). **e** Flow cytometry analysis of intracellular TCRβ expression in DN3 thymocytes from WT and *Arpp21⁻/⁻* mice displayed as representative histogram (upper panel) or statistical analysis (lower panel). Data are presented as mean values ± SD. *n* = 5, statistic significance was determined two-sided Student's *t* test ****P* < 0.001. **f** Semi-quantitative *P*CR analysis of Dβ2-to-Jβ2.6, Vβ8.2-to-DJβ2.6, Vβ11-to-DJβ2.6 and Vβ12-to-DJβ2.6 rearrangements in sorted CD25 + DN thymocytes from WT and *Arpp21⁻/⁻* mice. MEF cells were used as the negative control. CD14 was used as the loading control. Source data of (**a–f**) are provided as a Source Data file.

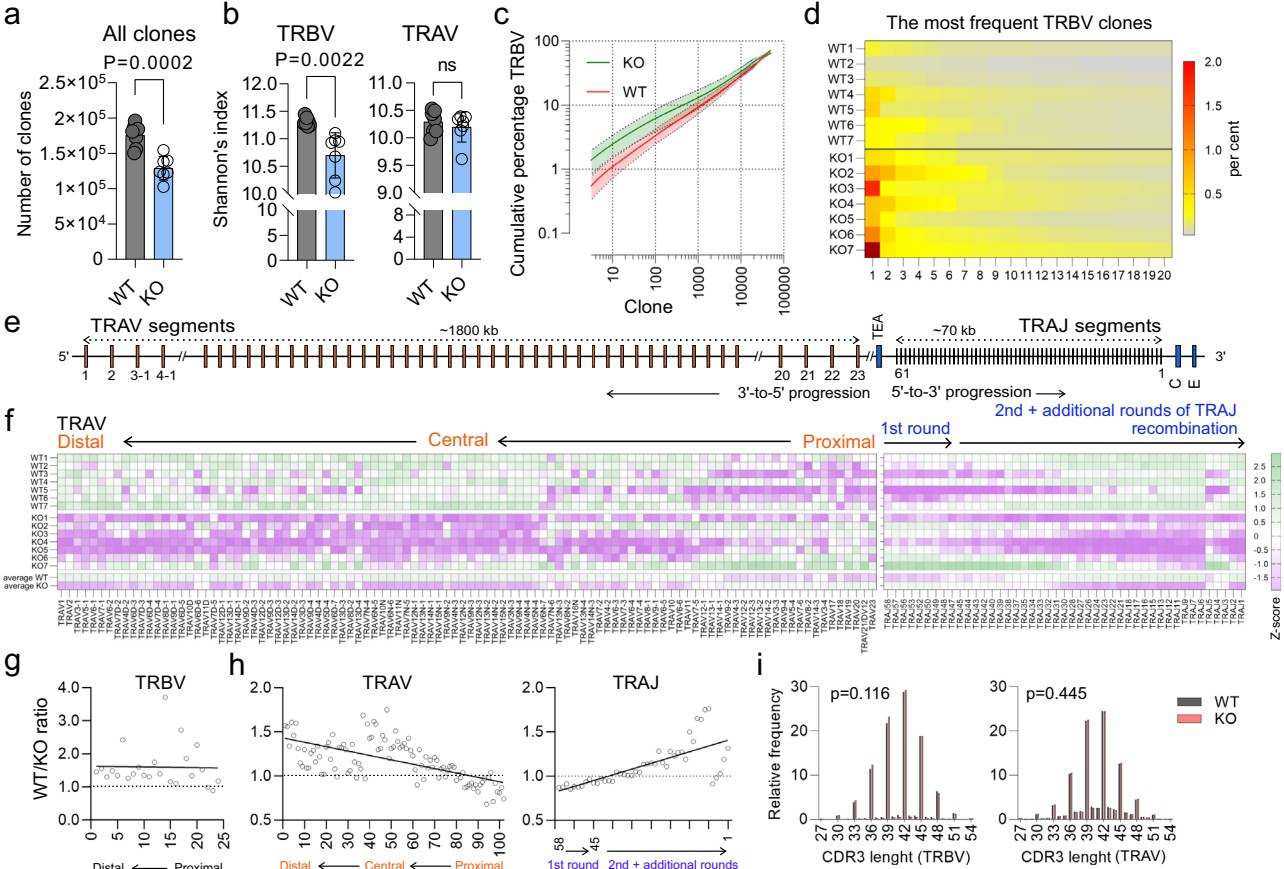

**Fig. 7 | Altered TCR repertoire in Arpp21-deficient T cells. a** The number of TRBV and TRAV clones identified in splenic CD4 T cells sufficient or deficient for Arpp21. **b** Repertoire diversity of TRBV and TRAV in peripheral CD4 T cells (**a, b**) error bars represent SD, significance was determined by unpaired *t* test, two-tailed. **c** Cumulative percentage of the most frequent 50,000 TRBV clones. Solid lines represent the mean of all analyzed mice, shaded areas indicate the standard deviation (**d**) Heat map showing the relative distribution of the top 20 TRBV clones. Each column represents a mouse with the indicated phenotype. **e** Schematic representation of TCR alpha locus, showing recombination order. Only TRAV and TRAJ elements are shown, TRDV elements have been omitted for clarity. Not to scale. Adapted after[57]. **f** Heat map showing the z-score of the relative usage of all

productive TRAV (left) and TRAJ (right) segments in WT and Arpp21-deficient splenic CD4 T cells. Each line represents a single mouse, except for the last two where the average for WT and KO is shown. WT vs. KO ratio of usage of all productive TRBV (**g**) as well as TRAV and TRAJ (**h**) segments in relation to their chromosomal location. **i** Length distribution (measured in nucleotides) of all TRBV and TRAJ CDR3 elements detected in WT and Arpp21 KO T cells, significance was analyzed by two-way ANOVA, effect of genotype is shown. Data are representative of one experiment, *n* = 7 for each genotype, each data point in (**a, b, d** and **f**) represents a single mouse. All analyzed mice are shown. Source data of (**a–h**) are provided as Source Data file.

and the Arpp21-3′-UTR interaction (*Rag1⁺/⁺* > *Rag1⁺/⁻* > *Arpp21⁻/⁻* = *Rag1*^3'del/3'del > *Rag1*^3'del/⁻). We quantified the Arpp21 effect and found a ~75% reduction in Rag1 signal in Western blots of thymocyte extracts from Arpp21 knockout mice, when comparing it to a titration curve of WT extracts (Fig. 9b).

To evaluate the global impact of such reduced Rag1 expression on thymopoiesis, we assessed thymic cellularity and found no major

differences between the genotypes (Supplementary Fig. 9a). To comprehensively analyze thymocyte development, we employed spectral cytometry, where we first looked for each major known cell type using 23 different parameters. After removing dead cells and contaminants (erythrocytes), we reduced 18 of the most relevant markers to two Uniform Manifold Approximation and Projection (UMAP) parameters. The manually gated thymic subpopulations were then plotted on the

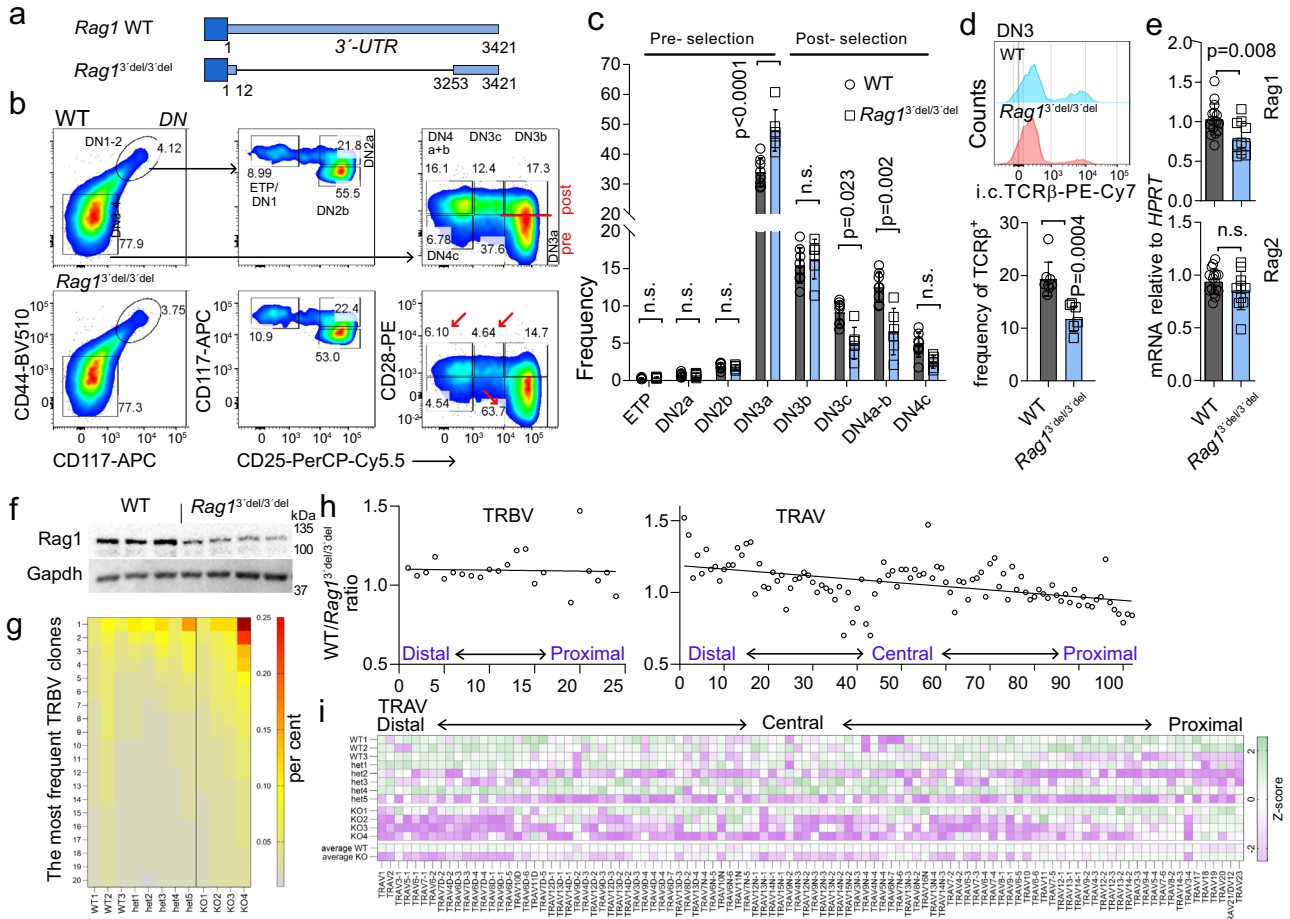

**Fig. 8 | Uncoupling Rag1 from Arpp21 regulation phenocopies Arpp21 deficiency. a** Schematic of the Rag1 3'-UTR locus showing the strategy used to generate *Rag1*[3'del/3'del] mice. **b** Representative plots of all major DN thymic progenitor populations in WT and *Rag1*[3'del/3'del] thymocytes. The red line indicates the point of beta selection, red arrows indicate the accumulation of pre-selected progenitors and the subsequent reduction of post-beta selected cells. **c** Quantification of the data shown in (**b**) (WT *n* = 8; KO = 6). Error bars represent SD significance was determined by two-way ANOVA followed by Sidak's multiple comparison test. **d** Representative histograms (top) and statistical analysis by unpaired *t* test, two-tailed (bottom) of the intracellular expression of the TCR beta chain in DN3 progenitors (WT *n* = 8; KO = 6). **e** RT-qPCR of *Rag1* and *Rag2* in unfractionated

thymocytes. **c**–**e** error bars represent SD. **f** Representative WB analysis of Rag1 expression in WT (*n* = 3) and *Rag1*[3'del/3'del] (*n* = 4) thymocytes. **g** Frequency of the 20 most common TRBV clones in peripheral CD4 T cells. **h** A plot showing the WT vs. KO (*Rag1*[3'del/3'del]) ratio of TRBV (left) and TRAV (right) in relation to their chromosomal location. **i** Heat map showing the z-score of the relative usage of all productive TRAV segments in WT and KO (*Rag1*[3'del/3'del]) splenic CD4 T cells. Each line represents a single mouse, except for the last two where the average for WT/het and KO (*Rag1*[3'del/3'del]) is shown. Data are representative of one experiment, *n* = 8 for combined WT and het mice, *n* = 4 for KO (*Rag1*[3'del/3'del]). Each data point in (**c**, **d** and **e**, **f**) represents a single mouse. All analyzed mice are shown. Source data for (**c**–**i**) and **f** are provided as a Source Data file.

UMAP 2-dimensional map (Fig. 9c) to show that UMAP discriminate all major known thymic subpopulations and roughly preserves the global relationship between them[33]. To further confirm this, we drew gates directly on the UMAP map to create 19 clusters around the areas with highest density of events and tested them for expression of all 18 markers used in the UMAP calculations (Fig. 9d). As shown in z-score map, the clusters created in this way corresponded well with the manual gates. For instance, C2 to C5 are cells in transition from DN3a to DN4c, C6, C7 and C9 correspond to DP, whereas C10 to C15 mark CD4 or CD8 single positive cells at different stages of development. Finally, C16 to C18 represent lineage-negative cells, C1 γδ T cells, with only the small C19 cluster being ill-defined.

Having established the UMAP projection as a viable method for assessing thymopoiesis, we compared all genotypes globally (Fig. 9e). This analysis revealed a conspicuous accumulation of cells around the β-selection point in all of the mutants as compared to wildtype mice (Fig. 9e). This prompted us to look more closely at lineage-depleted, CD25⁻ DN3-DN4 precursors where we confirmed this accumulation of pre-β-selected DN3a cells in all four mutant genotypes. The magnitude

of the block correlated well with the degree of Rag1 reduction in the various mutants (see Fig. 9a), with the strongest block observed in *Rag1*[3'del/-] thymocytes. DN3a cells constituted approximately 80% of all *Rag1*[3'del/-] DN3-DN4 cells versus 50.8% in WT, with a corresponding reduction of progenitors at DN4a+b and DN4 c stages (Fig. 9f, g and Supplementary Fig. 9c). Taken together, these data connect the genetic inactivation of Arpp21 as well as the deletion of the *Rag1* 3'-UTR to reduced expression of Rag1 protein and show how this reduction is correlated with an increasing block in thymocyte development. The data suggest that the more Rag1 expression is reduced, the greater the inhibitory impact becomes on the transition of DN3 progenitors across β-selection.

## Discussion

In this study, we provide evidence for a previously unrecognized post-transcriptional circuit that is required for optimal regulation of the Rag1 recombinase in thymocytes. Productive recombination of the T cell receptor is the defining goal of successful thymocyte development. As such, the expression of both proteins of the

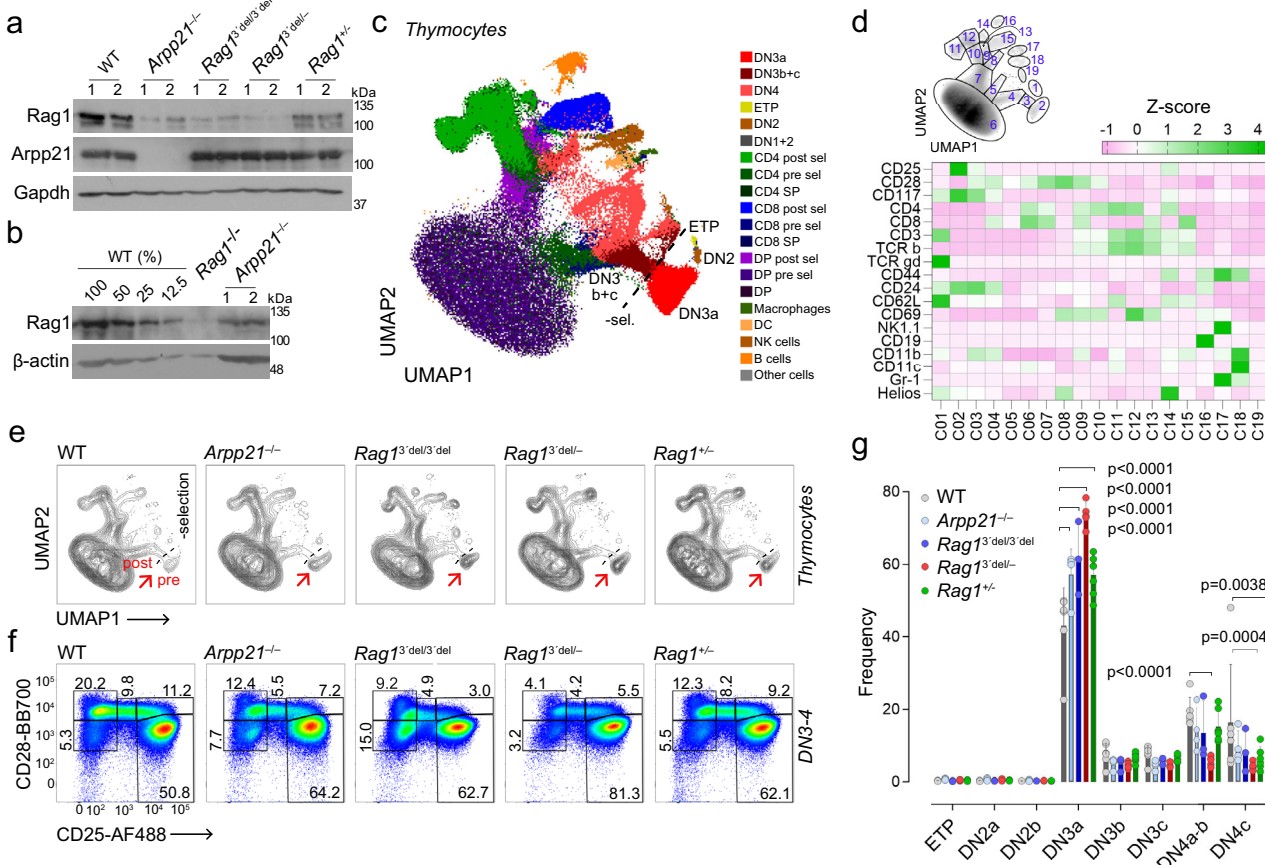

**Fig. 9 | The role of post-transcriptional regulation of Rag1 on thymocyte development. a, b** Immunoblot analysis of Rag1 and Arpp21 in thymocytes from two different control or mice with specified genotypes. Gapdh or β-actin serve as loading controls (*n* = 2 biological replicates). **c** Display of multicolor thymic staining data projected in two UMAP dimensions. UMAPs were calculated from concatenated files containing 250,000 live thymocytes from each mouse used in the experiment (6,250,000 cells in total). The overlay dot plot is colored according to manual gates. **d** Gates created around the cell clusters of the two UMAP dimensions. The heat map shows the expression of the 18 parameters used for UMAP in each cluster. **e** Representative UMAP projections of total thymocytes isolated from

mice of specified genotypes. Red arrows indicate the accumulation of progenitors before the β-selection point. **f** Representative pseudocolor plots of DN3-DN4 thymocytes isolated from thymi with the indicated genotypes. **g** Statistical analysis of all major DN thymic progenitors. The data are representative of two experiments, with two (**a**, **b**) or minimum three mice per each genotype (**c**–**g**). **c**–**e** All markers listed in (**d**) were used in the UMAP parameter calculations. **e** Each point represents a single thymus, means with SD are shown. Statistical analysis was performed using two-way ANOVA followed by Sidak's multiple comparison test. Source data for (**a**–**g**) are provided as a Source Data file.

recombination enzyme complex is subject to complex, multi-tiered regulatory mechanisms to ensure their precise temporal control. The induction of Rag1 and Rag2 expression in thymocytes initially depends on epigenetic and transcriptional mechanisms operating on highly conserved DNA *cis*-regulatory elements (CREs) that are recognized by the E protein transcription factors E2A and HEB[10]. The promoters of Rag1 and Rag2 and their shared T cell-specific enhancer region were found to be aligned within a defined three-dimensional chromatin topology. Apparently, this configuration promotes efficient binding by the E-proteins that induce stage-specific co-expression of Rag1 and Rag2 mRNAs[34–37]. Following stages with high Rag expression that enable TCR rearrangement, the shut-down of Rag1 and Rag2 protein expression is orchestrated in an interplay of different mechanisms: early work invoked PMA-induced destabilization of *Rag1* mRNA[38], and Rag1 protein degradation by the E3 ligase CRL4/VPRBP[39,40]. Moreover, cyclin A/CDK2−mediated phosphorylation and Skp2/SCF−dependent ubiquitination leads to degradation of Rag2 when thymocytes progress in the cell cycle[41,42]. Arpp21-mediated post-transcriptional enhancement of Rag1 expression introduces one more layer of regulation. For several reasons, we propose that the expression dynamics of Arpp21 and its regulatory activity have been adapted to perform this function. First, *ARPP21*

alleles originate from gene duplication at the time during evolution when the adaptive immune system developed (Supplementary Fig. 10)[14]. Second, the *Arpp21* gene giving rise to several isoforms of the mRNA as well as to an intron-contained miR-128-2 non-coding RNA, is most highly expressed in the thymus and the brain. The observed isoform expression reveals that the Arpp21 (100 kDa) protein with intact R3H/SUZ RNA-binding domain is the predominant isoform in the thymus, unlike the brain. In fact, this isoform is induced at the DN2/3 and DP stages of thymocyte development and is already present when the two waves of Rag1 mRNA transcription occur[11], and both Arpp21 and Rag1 mRNAs are downregulated at the DN4 stage. Third, Rag1 is a top-ranked mRNA target for Arpp21 RNA-binding activity as detected by iCLIP and among the strongest downregulated mRNAs in Arpp21−deficient thymocytes. Finally, we identified a regulatory feedback loop that modifies Arpp21 and switches its activity off when Rag1 activity is no longer needed. In these experiments, we found that SOCE through CRAC channels induced CaMK4-dependent phosphorylation, polyubiquitination and fast proteasomal degradation of Arpp21. Ca$^{2+}$ has recently been implicated in antigen receptor rearrangement through a genome-wide screen that identified *ATP2A2*[43]. *ATP2A2* encodes SERCA2, an ER membrane−localized Ca$^{2+}$−pump that maintains the ER Ca$^{2+}$

reservoir. SERCA pumps prevent ER store depletion, which would otherwise trigger CRAC channel activation and SOCE. Deletion of *Atp2a2* in mouse B cells decreased V(D)J recombination due to diminished Rag-mediated DNA cleavage, and combined inactivation of *Atp2a2* and its paralog *Atp2a3* caused increased cytosolic $Ca^{2+}$ levels, decreased Rag1 and Rag2 expression and a profound block of V(D)J recombination[43]. Conversely, the genetic inactivation of all three $IP_3$ receptor paralogs, which are required for store-depletion and SOCE, strongly affected thymocyte development by promoting DN to DP transition and caused aggressive acute lymphoblastic leukemia[44]. We have not observed a similar phenotype in *Stim1*$^{fl/fl}$;*Stim2*$^{fl/fl}$;*Vav-iCre* mice despite impaired degradation of Arpp21. This could be due to the possibility that failed degradation of Arpp21 in the absence of SOCE is not sufficient to induce malignant transformation of thymocytes, and/or the fact that Rag1 and Rag2 downregulation after TCR signaling occurs also independent of Arpp21. Interestingly, we determined downregulation of the Arpp21 protein during positive selection, but not in response to β-selection. One possible explanation is that $Ca^{2+}$ signals might be weaker in response to pre-TCR than TCR signals, despite ionomycin stimulation evoking similar $Ca^{2+}$ responses in DN and DP thymocytes. Another explanation could relate to differences in the expression of the factors that are involved in Arpp21 degradation, for example the so far unknown E3 ubiquitin ligase. Future work will be required to understand these differences.

Arpp21 is coexpressed with its Arpp21$^{short}$ isoform as well as with miR-128-2. Currently, we do not know whether Arpp21$^{short}$, which cannot bind to RNA, has a unique function, which could be related to the binding of interacting proteins[15]. The noncoding RNA miR-128-2 has been proposed to mediate antagonistic effects on targets shared with Arpp21[14]. We suggest that this antagonism may be less important in thymocytes prior to TCR rearrangement, since high Arpp21 expression may efficiently neutralize miR-128-dependent negative regulation. However, upon $Ca^{2+}$-induced degradation of Arpp21 protein, miR-128, which also has been reported to regulate the Arpp21 encoding mRNA[14,45], may add post-transcriptionally to the downregulation of the *Arpp21* mRNA. miR-128 could even contribute further, since the mouse *Rag1* 3′-UTR also contains putative miR-128 binding sites. One caveat, however, is that the *Arpp21* 3′-UTR−embedded binding site of miR-128 is conserved in amniotes, whereas the *Rag1* 3′-UTR binding sites of miR-128 are much less conserved.

The *Arpp21*$^{-/-}$ and *Rag1*$^{3del/3del}$ lines show phenotypes that are also observed in Omenn syndrome patients as well as in the previously generated hypomorphic Rag2$^{R229Q/R229Q}$ mutant[46]. However, *Arpp21*$^{-/-}$ and *Rag1*$^{3del/3del}$ mice differ from Rag2$^{R229Q/R229Q}$ mice, as they show no reduction in peripheral B cells and no signs of autoimmunity or lymphocyte infiltration in gut and skin, diarrhea, alopecia or erythrodermia[46]. These discrepancies may simply reflect that Arpp21 is not important to regulate Rag1 during B cell development. On the other hand, these mouse lines may also differ in Rag activity, since, different from Rag2$^{R229Q/R229Q}$ mice, peripheral T cell abundance is not affected in Arpp21−deficient mice. In the thymus the Rag complex introduces DNA double-strand breaks in the recombination signal sequences flanking the V, (D) and J fragments. This enzymatic activity occurs in progenitor cells with a very limited lifespan. If rearrangement is not successful, the cells die by neglect. A common phenotype of all three mouse models as well as Omenn syndrome patients is therefore oligoclonality of T cells[13,46]. One hallmark is the TRAV fragment usage, which reflects chromosomal location and Rag1/2 activity. Those TRAV fragments proximal to the J fragment are the first recombined, and their usage is relatively favored in cells with reduced levels or activity of Rag, and more distal fragments are underrepresented if recombination is inefficient. In fact, the shift towards proximal TRAV fragment usage is seen in Omenn syndrome patients[13,47] and similarly found in *Arpp21*$^{-/-}$ and *Rag1*$^{3del/3del}$ mice.

In these findings, an intricate network becomes apparent that ensures the robustness of early T cell development through Rag protein induction and guarantees subsequent downregulation of Rag proteins to prevent genetic instability. Arpp21 participates in this regulatory network. It promotes TCR rearrangement as a $Ca^{2+}$-sensitive post-transcriptional enhancer of Rag1 expression, increases TCR diversity and safeguards thymocyte development by being degraded in response to successful TCR rearrangement.

## Methods

### Mice

We confirm that our research complies with all relevant ethical regulations. All mice were on a C57BL/6 background. Mice of both sexes and different ages were used in a sex- and age-matched manner in our analyses. Mice were euthanized by cervical dislocation or exposure to $CO_2$. All animals were housed in a specific-pathogen-free barrier facility in accordance with the Helmholtz Zentrum München institutional, state and federal guidelines. All experimental procedures were performed following the rules and regulations approved by the local government (Regierung von Oberbayern reference no. 55.2-2532.Vet_02-19-68). *Arpp21*$^{-/-}$ mice were generated via CRISPR/Cas9-based gene editing by electroporation of one-cell embryos. For this, specific guide RNAs (Arpp21_In4_gRNA: 5′- GGATTTCAGTTCACCGTAAA-3′ and Arpp21_In5_gRNA: 5′- CCTGTTCTCAGTGACAAGCT-3′ or Rag1_UTR_gRNA1: GGCAAATACCAACTTCTATG and Rag1_UTR_gRNA2: TGAGGGGAGGTTTAGACACC) were used in form of in vitro transcribed single gRNA (EnGen® sgRNA Synthesis Kit, NEB, E3322). Prior to electroporation, the specific sgRNAs (200 ng/µl) were diluted in Opti-MEM buffer (Thermo Fisher Scientific) together with recombinant Cas9 protein (200 ng/µl, IDT) and incubated for 10 min at room temperature and 10 min at 37 °C to form the active ribonucleoprotein (RNP) complex. One-cell embryos were obtained by mating of C57BL/6N males (obtained from Charles River, Sulzbach, Germany) with C57BL/6N females super-ovulated with 5 units PMSG (Pregnant Mare's Serum Gonadotropin) and 5 units HCG (Human Chorionic Gonadotropin) and electroporated using the NEPA21 electroporator and a CUY501P1-1.5 electrode (Nepa Gene Co., Ltd., Japan). Zygotes were transferred into pseudo-pregnant CD1 female mice to obtain live pups. Gene editing events were analyzed using genomic DNA isolated from ear biopsies of founder mice and F1 progeny, using the Wizard Genomic DNA Purification Kit (Promega, A1120) following the manufacturer's instructions. *CamK4*$^{-/-}$ mice (Jax#:004994) and *Stim1/2*$^{fl/fl}$*Vav-iCre* mice (Jax#023350, Jax#023351, Jax#008610) were obtained from Jackson laboratory, and were described earlier[48,49].

### Orthogonal organic phase separation (OOPS)

The procedure was performed according to ref. 17. Isolate thymocytes and CD4$^+$ T cells from 6 to 8-week-old WT mice. Use $80 \times 10^6$ of thymocytes and $20 \times 10^6$ of CD4$^+$ T cells to do UV irradiation by Stratalinker 1800 at 254 nm three times (once at 0.4 J/cm$^2$ and twice at 0.2 J/cm$^2$) on ice. Harvest cells and wash them with PBS and resuspend the cell pellets in 1−2 ml of TRI Reagent (T9424, Sigma). Add Chloroform to isolate the interphase and repeat this step for four times and air-dry the protein-RNA complex at RT. Dissolve the pellet with TEAB buffer and then incubate with RNAse A, T1 (Thermo EN0551). Repeat the phase separation with TRI Reagent and Chloroform, and isolate protein from the organic phase. Prepare the protein lysate for MS analysis.

### Protein enrichment analysis of OOPS samples

The samples were diluted to 85 µl with 50 mM Tris-HCl pH 7.5. Reduction was performed by addition of 5 µl 100 mM dithiothreitol and 30 min incubation at 37 °C, followed by alkylation with 5 µl 400 mM iodoacetamide for 30 min at 24 °C. Proteins were purified with single-pot solid-phase-enhanced sample preparation (SP3) by washing away contaminants five times with 80% (v/v) ethanol[50].

Digestion was performed with 1.5 μg trypsin (Promega) overnight at 37 °C in the presence of 50 mM triethylammonium bicarbonate buffer pH 8.5. The resulting peptides were dried and resuspended in 30 μl 2% (v/v) acetonitrile (ACN), 0.05% (v/v) trifluoroacetic acid (TFA) of which 4 μl were used for LC-MS analysis.

### Crosslink site identification from cells

45 million irradiated thymocytes and 180 million irradiated CD4 T cells were resuspended in 6 M urea, 50 mM Tris-HCl pH 7.5 and disrupted by tip sonication with Sonifier SFX-250 (30 impulses 1 s/2 s on/off 10% amplitude, Branson Ultrasonics). The entire thymocyte lysate and half of the T cells lysate were further processed. Protein digestion was performed with trypsin (Promega) overnight at 37 °C in the presence of 1 M urea, 50 mM Tris-HCl pH 7.5 by addition of 100 μg enzyme for the thymocytes sample and 200 μg for T cells. RNA species were enriched with Direct-zol RNA Miniprep Plus kit (Zymo Research) according to manufacturer's protocol. The thymocytes and T cells samples were split in 6 and 8 spin-columns and combined again after elution. The pH was adjusted with 50 mM Tris-HCl pH 7.5 and urea was added to 1 M end concentration. To digest the RNA, 2 μl of RNAse I, 2 μl RNAse A and 0.5 μl of RNAse T1 (Thermo Scientific) were added to the samples and incubated overnight at 37 °C. Non-crosslinked nucleotides were depleted by reversed-phase chromatography with C18 SpinColumns (Harvard Apparatus) as previously described[19]. 10% of the eluate was taken for direct LC-MS analysis and the rest was subjected to TiO$_2$ enrichments with glycerol as competitor[19]. 10% of the eluate was separated for direct LC-MS analysis and the rest was subjected to basic reversed-phase chromatography (bRP) prefractionation with XBridge BEH C18 column (3.5 μm 1 mm × 150 mm, Waters): mobile phase A—10 mM ammonium hydroxide; mobile phase B – 80% (v/v) ACN, 10 mM ammonium hydroxide; flow rate—60 μl/min. The enriched crosslinks were separated with gradient from 5% mobile phase B to 10%, 42%, and 60% mobile phase B over 3, 34 and 8 min. The generated fractions were dried under vacuum and resuspended in 15 μl 2% (v/v) ACN, 0.05% (v/v) TFA and 4 μl were used for LC-MS analysis.

### Crosslink site identification of immunoprecipitated proteins

Arpp21-IP samples were purified with SP3 by washing away contaminants five times with 80% (v/v) ethanol. RNA digestion was performed with 2 μl of RNAse I, 2 μl RNAse A and 0.5 μl of RNAse T1 (Thermo Scientific) for 2 h at 37 °C in the presence of 50 mM Tris-HCl pH 7.5 and 1 M urea. Protein digestion was carried out with 3 μg trypsin (Promega) overnight at 37 °C. Depletion of non-crosslinked nucleotides and TIO$_2$ enrichment were performed as described above. The samples were resuspended in 12 μl 2% (v/v) ACN, 0.05% (v/v) TFA, and 5 μl were used for LC-MS analysis.

### LC-MS analysis

Chromatographic separation of peptides was achieved with Dionex Ultimate 3000 UHPLC (Thermo Fischer Scientific) coupled with in-house packed C18 column (ReproSil-Pur 120 C18-AQ, 3 μm particle size, 75 μm inner diameter, 33–35 cm length, Dr. Maisch GmbH). A multi-step gradient was formed with mobile phase A (0.1% (v/v) formic acid) and mobile phase B (80% (v/v) ACN, 0.08% (v/v) formic acid). For OOPS samples peptide elution was carried out with a gradient from 5% to 35% followed by an increase to 50% mobile phase B over 94 and 10 min. For global crosslink site identification experiments, a linear gradient was formed from 8% to 45% mobile phase B over 44 min for bRP fractions or 104 min for the rest of the samples. Peptides from IP experiments were eluted with a gradient from 5% to 42% mobile phase B over 43 min. Mass spectrometric analysis was carried out with Orbitrap Fusion Tribrid (Thermo Fischer Scientific) for IP experiments and Orbitrap Exploris 480 (Thermo Fischer Scientific) for the rest of the samples. Resolution was set to 60,000 (survey scans) and 15,000 (fragment scans) for OOPS samples and 120,000/30,000 for crosslink site identification experiments. Full details on additional acquisition parameters can be found in the provided raw files.

### Mass spectrometric data analysis

Protein enrichment MS raw files were analyzed with the MaxQuant software (version 2.1.4.0) with default settings except for the following modifications: label free-quantification and separate LFQ in parameter groups were enabled. Crosslinked and non-crosslinked samples were specified in separate parameter groups. For the database, UniProtKB/Swiss-Prot *Mus musculus* fasta file with 17090 sequences was accessed on 01.02.2022. MaxQuant protein groups output table was used for statistical analysis with Perseus (version 2.0.6.0). Reverse hits and proteins only observed with modified peptides were removed. LFQ intensities were log-transformed (base 2) and protein groups were filtered to have at least 3 valid values in at least one group. Missing values were imputed from normal distribution with width set to 0.3 and downshift set to 1.8. Significantly enriched protein groups were identified with two-sided Student's *t* test with permutation-based false discovery rate control at 5% and S0 parameter set to 2. The data have been deposited in the MassIVE database under the accession number.

Crosslink site identification was carried out with the OpenNuXL node of OpenMS (version 3.0.0). "RNA −UV(UGCA)" was selected as preset. RNA length was set to 3 and peptide length was limited from 5 to 30 amino acids for global crosslink site experiments. For IP samples the nucleotide length was limited to 2 and the minimum peptide length was set to 6. As database for the global search was provided the UniProtKB/Swiss-Prot *Mus musculus* fasta file with 17,090 sequences was accessed on 01.02.2022. The results at 1% spectral false discovery rate were used for following processing. For database of the immunoprecipitated proteins was provided the sequence of the respective protein and crosslink spectra were manually evaluated and validated.

### Protein expression and purification

Arpp21 recombinant proteins of the N-terminal subconstructs (engineered into the pETM11 vector containing TEV protease-cleavable N-terminal His-GB1 tag) were expressed in *E. coli* BL21(DE3) cells in LB or M9-minimal media (supplemented with 1 g/L $^{15}$N-ammonium chloride or 1 g/L $^{15}$N-ammonium chloride/2 g/L $^{13}$C-glucose) with 1 mM IPTG at the OD$_{600}$ ˜ 1.0 at 18 °C for 12 h. For purification, resuspended cells in 30 mM Tris/HCl pH 7.5, 500 mM NaCl, 10 mM imidazole were lysed using French press and sonication. The cleared lysates were applied to Ni-NTA column followed by washing (10 mM imidazole) and elution (500 mM imidazole) in the same initial buffer condition. To remove the tag, Arpp21-containing fractions were first incubated with TEV overnight at 4 °C, followed by repetition of Ni-NTA to remove TEV protease, tags, and uncleaved proteins. Next, the samples were further purified by ion exchange chromatography (HiTrap Q HP, Cytiva), in 20 mM TRIS pH8, NaCl (gradient 0.025–1 M), 2 mM DTT, and size-exclusion chromatography (HiLoad 16/60 Superdex 75 column, GE Healthcare), in the NMR buffer (20 mM sodium phosphate pH 6.5, 300 mM NaCl, 2 mM DTT).

### NMR spectroscopy

NMR experiments were recorded at 298 K on 1.2 GHz (Bruker Avance III HD, 3 mm cryo-TCI) and 600 MHz (Bruker AV III, cryo-TCI) spectrometers. NMR spectra were processed by TOPSPIN3.5 (Bruker) or NMRPipe[51], then analyzed using Sparky (T. D. Goddard and D. G. Kneller, SPARKY 3, University of California, San Francisco). All the samples were measured in the NMR buffer (above) with 10% D$_2$O for the lock signal. All the titrations were performed with 50 μM of $^{15}$N-Arpp21 samples by adding concentrated (2.5–10 mM) RNA oligonucleotides (U9, U6, A6, C6, G6—purchased from Dharmacon) to reach the indicated molar ratios. Backbone resonance assignments were obtained by analyzing standard triple resonance experiments HNCA, HNCACB and CBCA(CO)NH[52], measured using 250–300 μM $^{13}$C/$^{15}$N-Arpp21$^{130-260}$.

## ITC

ITC measurements (MicroCal PEAQ-ITC, Malvern Panalytical) were performed using non-isotopically labeled Arpp21[1-330] and Arpp21[61-260] and U9 RNA oligonucleotide (see above) in the NMR buffer with TCEP instead of DTT, at 25 °C. Arpp21 proteins (20 μM) in cell were titrated with U9 RNA (200 μM) in the syringe to reach the final molar ratio of (1:1.2).

## Western blotting

Cell lysis and SDS-PAGE were conducted using standard protocols described previously[20]. The following primary antibodies were used and diluted with 5% BSA: anti-Arpp21 (polyclonal 1:500, 11829-1-AP, Proteintech or monoclonal ((1:1000, 1E4[16]), anti-Roquin-1/2 (1:10, cl. 3F12, in-house production), anti-Rag1 (1:2000, ab172637, abcam), anti-Ubiquitin (1:1000, BML-PW8810-0100, Enzo Life Sciences), anti-Gapdh (1:10,000, CB1001, Merck Millipore), anti-β-actin (1:5000, A5316, Sigma) and anti-Tubulin (1:1000, sc-23948, Santa Cruz Biotechnology). The corresponding horseradish peroxidase (HRP)-conjugated secondary antibodies were anti-rat (1:3000, 7077, Cell Signaling), anti-mouse (1:3000, 7076, Cell Signaling), anti-rabbit (1:3000, 7074, Cell Signaling). Proteins were detected by Amersham ECL Prime Western Blotting Detection Reagent and FUSION FX (VILBER). Each Western blot was repeated at least two times.

## Isolation of thymocytes and splenocytes

Thymi and spleens were isolated from mice and lymphocyte suspension were prepared. To obtain purified DN thymocytes and DN1/DN2/DN3/DN4 cells, thymocytes were incubated with anti-CD4 (cl. RL1.72 in-house production) and anti-CD8 (cl. M31, in-house production), followed by Low-Tox-M rabbit complement lysis (CL3051, CEDARLANE) and deoxyribonuclease I (DN25-1G, Sigma-Aldrich) to deplete CD4- or CD8- positive thymocytes. Lympholyte®-M (CL5030, CEDARLANE) was then slowly added to the thymocyte suspension to obtain the lymphocyte layer after centrifugation at RT/2000G for 30 min. After centrifugation, interphase cells were collected, washed, stained and used for sorting. Thymocyte subsets were sorted on a FACS Aria Fusion sorter (BD)(Supplementary Fig. 11). Total CD4+ and CD8+ T cells from the spleen were isolated using the EasySep Mouse CD4+ T cell isolation kit (19852A, Stem Cell) and EasySep Mouse CD8+ T cell isolation kit (19853A, Stem Cell) according to the manufacturer's instructions.

## Ca²⁺ measurements by flow cytometry

Thymocytes from WT and *Stim1fl/fl; Stim2fl/fl Vav-iCre* mice were incubated with fluorescently labeled antibodies against CD45.2, CD4, and CD8 at room temperature for 15 min in the dark and loaded with 1 μM Indo1-AM (Molecular Probes, I1223) for 20 min at room temperature. At the beginning of each experiment, cells were kept in Ca²⁺-free Ringer solution (155 mM NaCl, 4.5 mM KCl, 3 mM MgCl₂, 10 mM D-glucose, 5 mM Na-HEPES, pH 7.4) followed by an addition of 1 μM ionomycin (Invitrogen, I-24222) to deplete ER Ca²⁺ stores. At the indicated time points, 2 mM Ca²⁺-containing Ringer solution (155 mM NaCl, 4.5 mM KCl, 2 mM CaCl₂, 1 mM MgCl₂, 10 mM D-glucose, 5 mM Na-Hepes, pH 7.4) was added into the tube to obtain a final Ca²⁺ concentration of 1 mM and to induce SOCE. Indo-1 fluorescence was measured at an emission wavelength of 396 nm and 496 nm after excitation at 355 nm. Samples were acquired on an LSR Fortessa flow cytometer (BD Biosciences) and analyzed using FlowJo software (TreeStar, versions 10.8.1).

## Isolation of B cell subsets

Isolate bones and squeeze them to get suspension bone marrow cells. Stain cells with biotin anti-CD11b and anti-Gr-1 and use streptavidin magnetic beads (Merck Millipore) to deplete myeloid cells. Harvest flow-through and stain cells with anti-CD11c-FITC, anti-CD19-PerCP-Cy5.5, anti-CD117-APC, anti-IgM-BV421, anti-IgD-BV650, anti-CD25-PE, anti-B220-PE-Cy7, and then wash cells twice with wash buffer. Sort B cell subsets on a FACS Aria Fusion sorter (BD). Lyse sorted B cell subsets in RLT lysis buffer and freeze them at −80 °C.

## Construction of plasmids

Mouse Arpp21 cDNA was amplified by PCR as described previously (14), and inserted into the retroviral plasmid for Tet-on expression with an N-terminal GFP fusion protein (pRetroX-tight, Clontech) using the In-Fusion HD cloning system or into the MSCV-Thy1.1 vector using the Gateway cloning system. Arpp21short cDNA was generated by Quikchange site-directed mutagenesis (Stratagene). The cDNAs of R3hdm1, R3hdm2 and R3hdm4 were cloned accordingly, and were PCR amplified from BC137772, BC117927 or BC095970, respectively. The 3′-UTR PCR products were amplified on C57BL/6 mouse cDNA and were inserted into the psiCHECK-2 vector (Promega) by In-Fusion technology (639649, Takara), the same as the cloning of psiCHECK-2-Rag1 3′-UTR mutations. All oligonucleotide sequences are shown in Supplementary Table 1.

## RNA isolation and RT−qPCR

RNA was isolated by column-based RNA isolation using RNeasy Mini Kits (74106, QIAGEN) or TRIzol (T9424, Sigma-Aldrich) and transcribed into complementary DNA using the Quantitect RT kit (205313, QIAGEN) according to the manufacturer's protocol. UPL Probe Library System (Roche) and Roche LightCycler 480 were utilized to quantify gene expression and the primers listed in Supplementary Table 2. TaqMan MicroRNA Reverse Transcription Kit (4366597, Thermo Fisher) was used for reverse transcription of miR-128 (002216), RNU6B (001093) and Sno-RNA202 (0011232). miR-128-3p and Sno-RNA202 qPCR were performed using TaqMan Universal PCR Master Mix (4304437, Thermo Fisher) as described[53].

## Tissue lysates

Dissected tissues from WT mice were placed in liquid nitrogen. Y cryoMill (Retsch) was used to homogenize the frozen tissues and the powders were kept at −80 °C. Each gram of powder was dissolved in 5 ml of cell lysis buffer.

## Detection of phosphorylation and ubiquitination

Thymocytes from WT mice were isolated and stimulated with 1 μM of ionomycin (407950-1MG, EMD Millipore) for 2 h with or without pre-incubation with 10 μM of proteasome inhibitor-MG132 (M7449-1ML, Sigma-Aldrich) for 90 min. When thymocytes were pretreated with BTP2 or KN93 for 90 min, the stimulation with ionomycin was shortened to 30 min. Whole thymocyte lysates were incubated with Arpp21 polyclonal antibody coupled to Protein A beads (10002D, Invitrogen) for 4 h at 4 °C and then washed five times. To analyze phosphorylation, beads were treated with Antarctic Phosphatase (AnP) (M0289S, NEB) for 20 min at 37 °C. The beads were then eluted with 1X SDS loading buffer for 5 min at 95 °C. The blot was probed with monoclonal anti-Arpp21. For the ubiquitination, the immunoprecipitated samples were blotted with anti-Ubiquitin.

## RNA-sequencing

DN2 and DN3 thymocytes from Arpp21-WT and Arpp21-KO mice were sorted to >97% purity and stored in RTL buffer (Qiagen) supplemented with fresh β-mercaptoethanol. RNA isolation, library preparation and sequencing were performed by Admera Health (https://www.admerahealth.com/). RNA quality was controlled using the Bioanalyzer 2100 Eukaryote Total RNA Nano (Agilent Technologies, CA, USA) and quantity was estimated using Qubit RNA HS assay (ThermoFisher). Low-input RNA poly-A selection libraries were prepared using the SmartSeq V4 with Nextera kit. Samples were sequenced on Illumina 2 × 150, with 20 million reads in each direction. Trimmed RNA-seq

reads were pseudo-aligned to the ENSEMBL GRCm38, Ensembl annotation version 101 transcriptome using kallisto, version 0.46. The data have been submitted to the Gene Expression Omnibus under the accession number GSE198798.

## TCR repertoire sequencing

Total RNA isolated from FACS-purified splenic CD4 T cells was processed using the SMARTer TCR a/b Profiling Kit (Takara). RNA was quality checked immediately after isolation and library preparation using Qubit and TapeStation and sequenced using the 600 cycle kit on the Illumina MiSeq platform. The generated data was demultiplexed and FastQC was performed. After trimming, the MIXCR pipeline was used to align reads to TCR clonotypes to assemble the final CDR3. CDR3 regions were defined by a stretch of amino acids between the N-terminal cysteine and C-terminal phenylalanine or tryptophan. Advanced analysis outputs were generated to compare VJ gene usage across samples and to determine Shannon's diversity and frequency changes in the repertoire across samples, as previously described[54,55]. The sequencing data is available at GEO site under the accession numbers GSE226368 and GSE226369.

The Shannon index was calculated with the following formula

$$H = -1 * \sum[(n/N) \times \ln(n/N)], \text{ where}$$

$n$−is the number of individuals of a given type (here TRAV or TRBV)

$N$−is the total number of individuals in a community (TRAV or TRBV).

## Individual-nucleotide resolution UV crosslinking and immuno-precipitation (iCLIP)

Thymocytes were isolated from mice and resuspended in 10 ml of cold PBS in a 15 cm diameter plate. Thymocytes were subjected to UV irradiation twice at 200 mJ/cm² at 254 nm in a Stratalinker 1800. After irradiation, thymocytes were washed and lysed in 1 ml of lysis buffer (50 mM Tris·HCl, pH 7.5, 100 mM NaCl, 1% (v/v) Igepal CA-630, 0.1% SDS, 0.5% sodium deoxycholate supplemented with 1 mM DTT and1x PI for 15 min on ice. Partial RNA digestion was performed by using RNase I (1:500 dilution, AM2295, Ambion) and Turbo DNase (AM2238, Ambion). RNA-protein complexes were immunoprecipitated with Arpp21 polyclonal antibody-coupled with 50 µl Protein A beads in 1 ml lysis buffer for 4 h at 4 °C. Subsequent processing and library preparation were performed as previously described[56]. Sequencing was performed on a Illumina HiSeq1500 sequencer with a length of 50 bp, single-read and 40 × 10⁶ reads/sample. Trimmed reads were mapped to the respective genome (assembly version GRCm38.p6 for all mouse samples) and its annotation (GENCODE release M25 for all mouse samples) using STAR (v2.7.6a). Reads were de-duplicated if they had identical UMIs. Peaks in WT samples with KO serving as background control were called using Homer (4.11) and the data were submitted to the Gene Expression Omnibus under the accession number GSE198798.

## DNA isolation and PCR assay for V(D)J recombination

Genomic DNA was extracted from sorted mouse CD25⁺ DN thymocytes (CD4⁻CD8⁻CD25⁺) or MEF cells using the QIAamp DNA Micro kit (56304, QIAGEN). Genomic DNA concentration was adjusted to 75, 15, and 3 ng and PCR amplification was performed using LongAmp Taq polymerase (M0323L, New England Biolabs) in the presence of Vβ and Jβ primers are listed in Supplementary Table 3. PCR products were separated on 1% agarose gels and exposed to UV light on a Quantum ST4 imaging system (Vilber Lourmat).

## Retroviral transduction

Retroviral particles were generated in HEK293T cells after calcium phosphate transfection using 50 µg of the appropriate plasmid and 5 µg of the pCL-Eco packaging vector (Addgene, 12371). After 48 h of cultivation, viral particles were filtered (0.45 µM) and mixed with 10 µg/ml polybrene (H9268, Sigma-Aldrich). Transduction in rtTA expressing MEF cells owas performed by spin infection (2 h, 32 °C, 300 rcf) and incubation for 6 h. Protein expression was induced in transduced cells by culturing in the presence of 1 µg/mL doxycycline (Dox) (A2951.0010, Applichem) for 16 h prior to flow cytometric analysis.

## 3′-UTR reporter assay

First, Hela cells were transduced with MSCV-IRES-Thy1.1 and MSCV-Arpp21-IRES-Thy1.1 and then selected using CD90.1/Thy1.1 MicroBeads (130-094-523, Miltenyi Biotec) to select HeLa cells overexpressing Arpp21. Selected lines were seeded into 24-well plates at 5 × 10⁴ cells per well. The following day, cells in the wells were transfected with 300 ng of psiCHECK2-Rag1 3′-UTR, psiCHECK2-Rag2 3′-UTR or empty psiCHECK2 using Lipofectamine 2000 (11668019, Thermo Fisher). After 24 h, cells were harvested for determination of luciferase activity using the Dual-Luciferase Reporter Assay System (E1910, Promega). Each transfection was analyzed in triplicate.

## Flow cytometry

Flow cytometry and cell sorting were performed on LSR Fortessa or CYTEK Aurora and FACSaria III (BD) cytometers, respectively. Monoclonal Abs specific for the following surface markers were used to define T and B cell populations: α-CD4 BV786 or BV421(RM4-5, 1:200), α-CD8 FITC or BV650 (53-6.7, 1:200), α-CD25 PE or PerCP-Cy5.5 (3C7, 1:200), α-CD28 PE (37.51, 1:50), α-CD44 BV421 or BV510 (IM7, 1:200), α-CD45R (B220) PE-Cy7 (RA3-6B2, 1:1000), α-CD117 APC (2B8, 1:100), α-IgM BV421 (RMM-1, 1:200) and α-TCRβ PE-Cy7 (H57-597, 1:1000). All Abs were purchased from BioLegend. For intracellular staining of TCRβ or IgM in thymocytes or bone marrow cells, the FoxP3/transcription factor staining buffer kit (00-5523-00, Invitrogen) was used according to the manufacturer's instructions. Briefly, thymocyte or bone marrow cells were first stained for CD4 BV510 (GK1.5, 1:100), CD8 APC/Cy7 (53.67, 1:200), CD44 APC (IM7, 1:200) and CD25 PerCP-Cy5.5 (PC61.5, 1:200) surface markers for 20 min at 4 °C, then fixed, permeabilized and stained with anti-TCRβ (H57-597, 1:1000) in permeabilization washing buffer for 30 min at 4 °C. Bone marrow cells were first stained for B220 BV605 (RA3-6B2, 1:200), CD19 BV711 (6D5, 1:200), IgM APC (II/41, 1:100), IgD PE-Dazzle 594 (11−26 c.2a, 1:800), CD11c FITC (N418, 1:400), CD117 APC (II/41, 1:100) and CD25 APC (II/41, 1:100) surface markers for 20 min at 4 °C, then fixed, permeabilized and stained Fab' fragments of FITC-goat anti−mouse IgM (115-097-020, 1:100) in permeabilization washing buffer for 30 min at 4 °C. Data were analyzed using FlowJo software (TreeStar) v10. For in depth analysis of DN stages, thymocytes were labeled with biotinylated anti-CD4 (1:200) and anti-CD8 (1:200) antibodies for 15 min, followed by 20 min incubation with anti-Biotin beads (Miltenyi Biotech, Cat#: 130-090-485). DP and SP thymocytes were depleted using LS+ columns (Miltenyi Biotech, Cat#: 130-042-401) according to manufacturers' instructions. All cells were stained with eF780 fixable viability dye for 20 min prior to surface staining. Surface staining was performed in FACS Buffer (2% FCS, 1 mM EDTA in PBS) supplemented with Brilliant Staining Buffer (Thermo Fisher, Cat#: 00-4409-42) for 20 min on ice. Cells were fixed and permeabilized using the eBioscience™ Foxp3/Transcription Factor Staining Buffer Set (Thermo Fisher) following the manufacturers' instructions. Intracellular staining was performed in PermBuffer at 4 °C.

For spectral flow cytometry analysis of thymus and bone marrow following antibodies were used. *Thymus panel:* α-CD28 BB700 (37.51, 1:100), α-TCRγδ APC (eBioGL3, 1:200), α-CD24 BV605 (M1/69, 1:800), α-CD69 PE-Cy7 (H1.2F3, 1:400), α-CD11c BUV737 (N418, 1:200), α-CD62L eF450 (MEL-14, 1:800), α-Gr-1 SB645 (RB6-8C5, 1:200), α-CD25 Alexa Fluor 488 (PC61.5, 1:100), α-TCRβ BUV395 (H57-597, 1:400), α-Ter-119 BUV496 (TER-119, 1:100), α-NK1.1 BUV563 (PK136, 1:400), α-CD4 BUV661 (GK1.5, 1:200), α-CD117 BV421 (QA17A09, 1:100), α-CD11b

BV480 (M1/70, 1:400), α-CD44 BV570 (IM7, 1:400), α-CD19 BV711 (6D5, 1:800), α-CD8 BV750 (53-6.7, 1:400), α-CD3 APC-Fire 810 (17A2, 1:400), and α-Helios PE (22F6, 1:400).

### Generation of antibodies

For the generation of the 8G2 anti-Arpp21 monoclonal antibody, rats were immunized with a recombinant Arpp21 (aa 1-330) protein fragment. Antibodies from hybridoma supernatants were screened by flow cytometry for clones that recognized bead-coupled protein (IntellCyt iQue Screener; Sartorius). Candidates were validated by testing recognition of ectopic expression of Arpp21 but not Arpp21$^{short}$ in transfected HEK293T cells. Experiments were performed with 8G2 monoclonal hybridoma supernatant (rat IgG2a/k) (1:10 dilution).

### Statistical analysis

All statistical analysis for flow cytometric data was performed with GraphPad 7.0 software using Student's $t$ test or one-way or two-way ANOVA as indicated.

### Reporting summary

Further information on research design is available in the Nature Portfolio Reporting Summary linked to this article.

## Data availability

All data that support the findings of this study are available in the article and its Supplementary files or from the corresponding authors upon request. The RNA-Seq and Arpp21-CLIP data generated in this study have been deposited in the Gene Expression Omnibus database under the accession number GSE198798. The TCR sequencing data are also available at the GEO site under the accession numbers GSE226368 and GSE226369. Proteomic data generated in this study have been deposited in the MassIVE database under the accession number MSV000091606. Source data are provided with this paper.

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

## Acknowledgements

We would like to thank Lena Esser (Ludwig-Maximilians-Universität München), Miki Jishage (New York University), and Ralf Pflanz and Monika Raabe (Max Planck Institute for Multidisciplinary Sciences) for excellent technical assistance. M.X. received a scholarship from the China Postdoctoral Council program of the Helmholtz Association (OCPC). The work was supported by the German Research Foundation grants SPP 1738 #255046332 (to F.G.W.) and SPP 1935 #273941853 (to V.H., M.S., J. König and H.U.), SFB-TRR 338/12021-452881907 (project C02 to V.H.), SFB 1054 # 210592381 (project A03 to V.H. and project B11 to C.D.), SFB-TRR 355 #490846870 (project A02 to C.D. and project A06 to V.H.) as well as HE3359/7-1 (#432656284 to V.H.), HE3359/8-1 (#444891219 to V.H.), and DFG LY 150/2-1 (AOBJ 690849 to M.Ł.) as well as grants from the Wilhelm Sander (#2018.082.2 to V.H.) and Krebshilfe (#70113538 to V.H.) Foundations and PHS NIH RO1 AI136924 (to G.C.T.).

## Author contributions

V.H., F.G.W., and M.Ł. conceived and designed the project with input from J. Kisielow. V.H. and M.Ł. wrote the manuscript with contributions from F.G.W., T.I.-K. and M.X. M.X. and K.P.H. were involved in OOPS and qPCR analyses. A.C. and H.U. performed and analyzed mass spectrometry experiments. M.X., Y.W., W.P., C.H., B.A., K.P.H., S.F., and G.C.T. contributed immunoblotting experiments. H.-S.K. and M.S. conducted protein expression, purification and performed and analyzed NMR spectroscopy and ITC. R.F. and V.H. generated antibodies. T.I.-K. and M.X. performed iCLIP experiments with the help from J. König. T.S. and T.I.-.K. analyzed iCLIP and RNA-Sequencing data. F.G. and W.W. generated Arpp21–/– and Rag1³ᵈᵉˡ/³ᵈᵉˡ mice and Rag1⁻/³ᵈᵉˡ mice with the help of C.D. M.Ł., M.X., T.I.-K., T.R., and N.Z. conducted thymocyte and B cell flow cytometry. M.Ł. performed TCR repertoire analyses and M.X. the V(D)J recombination analyses.

## Funding

## Competing interests

The authors declare no competing interests.

## Additional information

[1]Research Unit Molecular Immune Regulation, Molecular Targets and Therapeutics Center, Helmholtz Zentrum München, Munich, Germany. [2]Department of Integrated Traditional Chinese and Western Medicine, Union Hospital, Tongji Medical College, Huazhong University of Science and Technology, Wuhan, China. [3]Institute for Immunology, Biomedical Center (BMC), Faculty of Medicine, Ludwig-Maximilians-Universität in Munich, Planegg-Martinsried, Germany. [4]Institute of Structural Biology, Molecular Targets and Therapeutics Center, Helmholtz Zentrum München, Neuherberg, Germany. [5]Technical University of Munich, TUM School of Natural Sciences, Department of Bioscience and Bavarian NMR Center (BNMRZ), Garching, Germany. [6]Max Planck Institute for Multidisciplinary Sciences, Bioanalytical Mass Spectrometry, Göttingen, Germany. [7]Institute of Developmental Genetics, Helmholtz Zentrum München, Neuherberg, Germany. [8]Department of Pathology, New York University, Grossman School of Medicine, New York, NY, USA. [9]Department of Medicine, Beth Israel Deaconess Medical Center, Harvard Medical School, Boston, MA, USA. [10]Institute for Molecular Biology, Biomedical Center (BMC), Faculty of Medicine, Ludwig-Maximilians-Universität in Munich, Planegg-Martinsried, Germany. [11]Monoclonal Antibody Core Facility, German Research Center for Environmental Health, Neuherberg, Germany. [12]Research Unit Type 1 Diabetes Immunology, Helmholtz Diabetes Center at Helmholtz Zentrum München, Neuherberg, Germany. [13]German Center for Diabetes Research (DZD), Neuherberg, Germany. [14]Division of Clinical Pharmacology, Department of Medicine IV, Ludwig-Maximilians-Universität München, Munich, Germany. [15]Max von Pettenkofer Institute, Faculty of Medicine, Ludwig-Maximilians-Universität in Munich, Munich, Germany. [16]Institute of Molecular Biology (IMB), Mainz, Germany. [17]Chair of Developmental Genetics, Munich School of Life Sciences Weihenstephan, Technical University of Munich, Freising, Germany. [18]Munich Cluster of Systems Neurology (SyNergy), Munich, Germany. [19]German Center for Neurodegenerative Diseases (DZNE) site Munich, Munich, Germany. [20]University Medical Center Göttingen, Department of Clinical Chemistry, Bioanalytics Group, Göttingen, Germany. [21]Göttingen Center for Molecular Biosciences, Georg-August University Göttingen, Göttingen, Germany. [22]Cluster of Excellence 'Multiscale Bioimaging: from Molecular Machines to Networks of Excitable Cells' (MBExC), University of Göttingen, Göttingen, Germany. [23]Institute for Molecular Health Sciences, ETH Zürich, Zürich, Switzerland. [24]Institute for Integrative Neuroanatomie, Charite-Universitätsmedizin Berlin, Corporate Member of Freie Universität Berlin, Humboldt-Universität zu Berlin, and Berlin Institute of Health, Berlin, Germany. [25]Department of Pediatrics and Adolescent Medicine, University Medical Center Ulm, Ulm, Germany. [26]Present address: Cancer Immunology and Immune Modulation, Boehringer Ingelheim Pharma GmbH & Co. KG, Biberach an der Riss, Germany. [27]Present address: Repertoire Immune Medicines (Switzerland) AG, Schlieren, Switzerland. [28]These authors contributed equally: Meng Xu, Taku Ito-Kureha. [29]These authors jointly supervised this work: Marcin Łyszkiewicz, Vigo Heissmeyer. ✉e-mail: jk@repertoire.com; gregory.wulczyn@charite.de; marcin.lyszkiewicz@uni-ulm.de; vigo.heissmeyer@med.uni-muenchen.de

