## [Peer Review File · Nature Communications]

The thymocyte-specific RNA-binding protein Arpp21 provides TCR repertoire diversity by binding to the 3'-UTR and promoting Rag1 mRNA expressionREVIEWER COMMENTS

Reviewer #1 (Remarks to the Author):

This manuscript begins with an experiment which identifies RNA binding proteins in the mouse thymus using the OOPS method. By comparison with a previous dataset from the same lab it identifies hundreds of RNA binding proteins that appear to be absent from peripheral CD4 T cells. This difference is likely in part explained by the more complex mixture of cell types in the thymus.

The manuscript then focusses on the regulation and function of Arpp21. This choice is based in part on the results of a previous study from 2001 by one of the authors of the submitted manuscript which identified a protein (called TARRP) that is expressed in thymocytes around the time of commitment to T cell lineage. This manuscript reported TARRP expression is switched off as a consequence of TCR engagement during positive selection.

The current manuscript refines these observations by reporting mRNA analysis and by demonstrating that TCR dependent calcium signalling promotes the loss of the protein via ubiquitination and degradation.

A key novel experiment is the modification of the ARPP21 locus to create a null allele and from there to assess the effect of loss of function on lymphocyte phenotype. This is challenging task because the locus also harbors a microRNA (miR-128-2). The authors claim that there is no significant effect of their mutation on the expression of miR-128-2. This is based on the PCR analysis of whole thymus. However, while not statistically significant (NB I could not find any indication of what the statistical analysis was), miR-128-2 expression shows a strong decreasing trend upon Arpp21 KO. Because of the cell population sampled, it is not ruled out that miR-128-2 is more strongly affected in DN3 or DP thymocytes which express the greater amounts of host transcript.

A previous study noted a substantial overlap between transcripts with binding sites for miR-128 and ARPP21, with common targets but opposing functions of ARPP21 and miR128. This complex interaction and uncertainty about whether the amounts of miR-128-2 in DN3 or DP

cells might be different, and therefore influencing gene expression, needs further clarification. This is important for the mechanistic understanding of thymic development because whether an ARPP21/MiR182 axis is biologically relevant in thymus as in neuronal development (REF 14) is of interest and relevant to the current manuscript.

It is striking that the modification of the Arpp21 locus causes a strong increase of the short ARPP21 protein isoform (Fig 3B), which was barely detectable in the WT thymus. Although the short protein doesn't have RNA binding domains, it is possible that this protein contributes to the phenotype reported here and this is not addressed. Is the mRNA encoding the short isoform, like that of the long isoform, a target of miR-128-2?

The NMR studies in Figure 4 and supplemental 5 reveal a domain-intrinsic specificity for uridine-rich sequences. This is consistent with the results of the CLIP in this manuscript and with a published study (cited by the authors as Ref-14) which concluded ARPP21 recognition of uridine-rich sequences with high specificity for 3'UTR. The ability of R3H domains to bind RNA is known and the experiments shown here confirm this is also the case for ARPP21.

The manuscript reports the results of ARPP21 CLIP on thymocytes which identifies 6779 direct targets. The analysis intersects these with genes that are differential expressed in an RNA.seq experiment comparing DN2/3 thymocytes from KO and WT. From here the focus shifts to the regulation of the Rag1 mRNA which encodes a protein essential for VDJ recombination.

The authors show that Rag1 mRNA contains a 3421 nucleotide long 3'UTR and map out the functional interactions using a reporter system which are extensive and consistent with the profile of RNA binding determined by CLIP. They then generate a mouse in which the majority of the Rag1 3'UTR is removed. This approach can be challenging to interpret as the interactions between the rag1 3'UTR with other factors are also lost. While the phenotype presented appears similar to the ARPP21 KO, more specific UTR mutaitons that only blocked only the ARPP21/rag1 UTR interaction would be stronger evidence of a causal relationship.

The most novel aspect of this manuscript are the claims based around the thymic phenotype of the ARPP21 and Rag3' UTR mutant mice. The phenotype presented in Figures 5 A-C and 6A-D appears modest and more information is needed to clarify and strengthen the case for the interpretation the authors make. It will help to show cell numbers and to be clear about the characteristics of the mice being compared. The age and sex as well as whether littermates are being compared is relevant, it might also be helpful to have a quantitative analysis of cell numbers in heterozygous mice.

The results with early B cell development reports a small effect on the frequency of preB cells, but the reported results do not include the detection of cytoplasmic IgH which is a key relevant marker for these cells and this phenotype. This analysis would also benefit if cell numbers were presented.

The effects on Rag1 protein amounts are estimated by Western blot of whole thymus (Fig 6B and 8F) are variable between the ARPP21 and Rag3' UTR mouse models and there is some inconsistency between the cell types measured by in the different models. It is not clear if reduced amounts of Rag1 protein account for the phenotype reported. Given the reduced amounts in 6b are about half of the wildtype, one can predict Rag heterozygous mice should have a similar phenotype. Its not clear if that has ever been looked at.

It is unclear how a reduction in the amount of RAG1 leads to reduced clonotype diversity. Once a Rag induced break is made, the N additions etc that introduce junctional diversity shouldn't be impacted. Is it possible that ARPP21 has some impact on this process?

In the Arpp21 deficient mice there is a phenotype at the beta selection checkpoint but not at the double positive stage in terms of cell numbers. The authors argue that premature downregulation of RAG1 leads to reduced recombination mainly of the beta chain. The relevance of the expression of Arpp21 in DP thymocytes remains uncertain and it is not clear why the reduced RAG1 expression in DP thymocytes does not lead to detectable changes in Valpha chain rearrangement.

The repertoire analysis for both mouse strains is performed on splenic CD4+ T cells. These data include 100- to 200-K sequences in each sample, giving a lot of statistical power, able, in principle, to detect small differences robustly. It is surprising that the effects of the Rag1 UTR mutation are smaller than those of Arpp21KO in terms of the effect on the clones. This is an indication the phenotype of the Arpp21 KO may result from additional effects). The split axis in 7b is confusing, it is not clear why it is included or needed, particularly as it's split differently for TCRVB on the left and TCRVA on the right. I could not find, in the figure, or in the legend, whether this is a linear or a log scale. Nor was it clear what methodology was used to determine the values for the Shannon index in the graphs. There are two papers cited in the methods with no further explanation of which or how the tools were used. How does the analysis take into account the lower number of sequences in the Arpp21 ko?

The authors suggest that reduced Rag1 might decrease the efficiency of recombination and the VDJ/DJ ratio, (there is evidence for a reduced V-DJ with normal DJ on a gel in 6f), and could decrease the number of V(D)J clonotypes in a population, which does occur in the Arpp21 ko. It is unclear if the reduction in Rag1 can account for this.

It is unclear why there is a difference between TCRB and TCRA and what this means.

They mention that for TCRA, proximal V genes are used more when Rag expression is low, and their data agrees with this (7e, 7f). So that shifts the distribution of V genes used, but doesn't appear to impact on diversity, since in 7b, there is no difference in Shannon index for TCRA. The current presentation is thus rather unclear about TCR repertoire diversity per se and not convincing that this is more than reduced recombination. A role of Arpp21 in controlling positive or negative selection in the thymus or additional function of the short isoform could be relevant and needs to be acknowledged when comparing these two mouse models as the phenotypes are clearly different.

Overall, this is of interest because it appears to be a new mechanism that regulates Rag1, but the title is misleading, overstating the effects on TCR diversity – and suggesting that these are due to reduced Rag expression. Reduced diversity could also be due to reduction

in one of the many other Arpp21 targets, or other changes as the RAG1 UTR models shows a less strong phenotype.

There are some interesting parallels as the authors discuss with the Omenn syndrome and potential oligoclonality and the usage of more proximal TRAVa segments due to lower functionality of Rag. However, the two mouse models introduced here show no evidence of any T cell dysfunction, self-reactivity and the mice are healthy. Has the T cell biology been investigated? Is there a loss of T cell function/dysfunction? If none of this is the case then these mice might not be a good model for Omenn syndrome.

Minor comments

Line 367 indicates that the Rag13⁻-UTR-KO mice were generated by crossing of two founders to generate homozygous F1 offspring. Have unanticipated mutations in the rag1 locus introduced by cas9 been ruled out?

Improve figure legends and methods. Particularly, include keys for colours and symbols in graphs and include the number of biological replicates when missing. Authors only show one representative western blot picture for each experiment; they could include a graph showing relative protein levels for all the biological replicates done in an experiment.

Lines 136-138 and Fig 1C: where do the 490 proteins come from?

Fig 2C/2D: western blot pictures are cropped to close to the Arpp21 band, and it is not possible to see if there is the phosphorylated band in these blots.

Fig 2F: there are no bands at 100 kDa in the ubiquitin blot following Arpp21 IP. There should

be a band at the height of the protein immunoprecipitated. There is a faint signal just below 135 KDa. It would help if the authors point to the bands of interest with arrows.

Figure 4 e misses the 1 in Arpp21

Suppl. Fig 2b: there is an accumulation of the phosphorylated Arpp21 protein also upon Iono treatment which was never seen before; in fact, the phosphorylated band appeared upon Iono treatment and proteasome inhibition.

Correct Spelling in supplemental 6b

Reviewer #2 (Remarks to the Author):

The noteworthy results of this manuscript are the discoveries that the RNA binding protein Arpp21 associates with the 3'UTR of Rag1, promoting its stabilization and role in TCR rearrangement. Loss of Arpp21, or removing the binding region in Rag1 3'UTR, results in a less diverse TCR repertoire. The methodologies employed are sound. On the whole, major claims (particularly those made early in the manuscript) are well-supported by the data. However, some minor weaknesses and points of confusion exist and should be addressed:

1. (Figure 5b and c) Authors claim that while overall numbers of developing T cells in the thymus are unchanged, there are reductions of cells at DN3 stage interpreted as a developmental block during beta chain rearrangement.
 - a. Given that there is also an increase of cells at DN4 stage in the KO, it may also be interpreted that cells move through the DN3 stage more quickly and that the block may be later at the DN4 stage when pre-TCRb signal is required to progress. Is there evidence that more T cells are dying for lack of a functional pre-TCR signal?
 - b. The differences in frequency are quite subtle and it may help to further interrogate the phenotype or function of thymic emigrants to demonstrate the biological significance of Arpp21 loss on T cell development.
2. (Figures 7 and 8) It is interesting that there are differences between TRBV and TRAV gene rearrangement in absence of Arpp21, namely, TRBV diversity is reduced while TRAV

diversity is unchanged, and TRBV gene usage is unchanged while TRAV gene usage is skewed.

a. Are these observations related to the fact that TRB alleles are sequentially rearranged, while TRA alleles rearrange simultaneously (and are sometimes co-expressed)?

b. Diversity was not impacted by disrupting Rag1 3'UTR interaction with Arpp21 but gene usage skewing still exists. Might this suggest that the observed impact of Arpp21 loss on diversity or gene usage skewing is Rag1-independent?

Overall, the data that Arpp21 and Rag1 interact appear strong, but the functional consequences of this still remain a bit of a mystery.

Reviewer #3 (Remarks to the Author):

RNA binding proteins play complex and essential roles in gene regulation, but their functions are poorly understood. Xu and colleagues reveal the function of a known binding protein - Arpp21, which they show plays an important and previously unappreciated role in regulation of the Rag1 recombinase that rearranges T cell receptor genes during development in the thymus. The authors construct elegant Arpp21 deficient mice that leave the short isoform of Arpp21 intact, which is important as Arpp21short is important in neural development. These mice are shown to have a defect in TCR beta gene rearrangement and also in a process known as sequential TCR alpha gene rearrangement, which requires high persistent levels of RAG activity and is important to maximize the breadth of the TCR repertoire in T cells. The authors map the binding sites of Arpp21 in Rag1. They make a further elegant mouse mutant in the 3' site of RAG1 that binds Arpp21, and establish that this RAG1 3' UTR mutant phenocopies the Arpp21 mutant they characterized in its effects on RAG1 expression.

The work here represents an exceptionally thorough set of experiments. The experiments are well performed, the writing is clear and concise. One minor comment is that a similar scheme of Arpp21 and Arpp21short added to Figure 1 would help the reader. Outside this detail which the authors are free to ignore, I really have no substantive criticisms or suggestions for the authors. I wish to congratulate them on their interesting, thorough and

well-performed study.

[redacted]

[Editorial Note: reviewer name is redacted as per their wishes]

We would like to thank all reviewers for their comments and their constructive criticism, which helped to improve our paper. We would also like to personally thank [redacted] [**editorial note: reviewer name redacted**], for expressing [their] enthusiasm for this manuscript under review.

In the revised version we have now addressed all comments with new experimentation or with additions or clarifications in the main text or in our point-by-point response. The main changes to our manuscript are:

1. We have reanalyzed miR-128 expression in the main subsets of thymocyte development in WT and Arpp21^{-/-} mice (**Extended data Figure E3, new subfigure 3b**).
2. We have investigated the enrichment of binding sites for miR-128 (and all other miRNAs) in the Arpp21-bound transcriptome in comparison to their presence in non-bound mRNAs of thymocytes. Please find the new data in the **Extended data Figure E3, new subfigure 3d**.
3. We have re-investigated B cell development in WT and Arpp21^{-/-} mice as requested with intracellular IgM staining (**new Extended data Figure 7, new subfigure E7f**).
4. We have newly generated and utilized a monoclonal antibody (8G2) that is specific for full-length Arpp21 but does not recognize Arpp21^{short} (and does not recognize R3hdm1, R3hdm2 and R3hdm4) as validated in the **new Extended data Figure E6**. We have utilized this novel tool to further analyze the regulation of the Arpp21 protein levels during thymocyte development and pre-TCR signaling or positive selection as shown in the **main Figure 5, new subfigure 5c**.
5. Most importantly, we have set up new mouse breedings and generated several new genotypes i.e. WT, Arpp21^{-/-}, Rag1^{3'del/3'del}, Rag1^{3'del/-} and Rag1^{+/-} or Rag1^{-/-}, which we analyzed in comparison in the **new main Figure 9** and **new Extended data Figure E9** for their level of Rag1 protein expression and thymocyte phenotypes.

REVIEWER COMMENTS

Reviewer #1 (Remarks to the Author):

This manuscript begins with an experiment which identifies RNA binding proteins in the mouse thymus using the OOPS method. By comparison with a previous dataset from the same lab it identifies hundreds of RNA binding proteins that appear to be absent from peripheral CD4 T cells. This difference is likely in part explained by the more complex mixture of cell types in the thymus.

We agree with the assessment as our biochemical start-point of identifying thymus-specific RBPs required large cell numbers that cannot be obtained at highest purity. Nevertheless, thymocytes are up to 85% double-positive cells, which may for the majority of detected RBPs reflect their expression and binding to RNA in DP cells. Knowing the limitations of the comparison (activated peripheral CD4+ and ex vivo isolated thymocytes) we did not overstate global differences but went after one truly thymus specific RBP.

The manuscript then focusses on the regulation and function of Arpp21. This choice is based in part on the results of a previous study from 2001 by one of the authors of the submitted manuscript which identified a protein (called TARRP) that is expressed in thymocytes around the time of commitment to T cell lineage. This manuscript reported TARRP expression is switched off as a consequence of TCR engagement during positive selection. The current manuscript refines these observations by reporting mRNA analysis and by demonstrating that TCR dependent calcium signalling promotes the loss of the protein via ubiquitination and degradation.

We agree that we have very much refined the initial observation. Please note, that the previously reported finding that positive selection in thymocytes correlates with reduced Arpp21 expression did not discriminate between Arpp21 long and short proteins and lacked a mechanistic explanation. We demonstrated degradation of Arpp21 as a consequence of Ca²⁺-dependent phosphorylation and poly-ubiquitination, for which we identified the critical molecular players ORAI/STIM and CaMK4 as well as the proteasome using both inhibitors as well as knockout mouse lines. Also, another refinement became possible during the preparation of this revision, since we have now generated a new monoclonal antibody that only recognizes the long (RNA-binding) isoform of Arpp21 and is suitable for flow cytometry. With this new tool we were able to assign pre-, ongoing- and post-positive selection markers with high, intermediate and low Arpp21 protein levels, fully supporting the physiologic relevance of the newly identified signal transduction pathway in positive selection. However, to our surprise, we did not detect degradation of Arpp21 at stages of pre-TCR signal transduction, although double-negative thymocytes are also capable of generating a similar Stim1/Stim2 dependent Ca²⁺ influx, if stimulated via ionomycin (**Extended data Fig.2d**). The data are depicted in the new **Figure 5c**, and we added a discussion paragraph to this revised version: " Interestingly, we determined downregulation of Arpp21 during positive selection, but not in response to β -selection. One possible explanation is that Ca²⁺ signals might be weaker in response to pre-TCR than TCR signals, despite ionomycin stimulation evoking similar Ca²⁺ responses in DN and DP thymocytes. Another explanation could relate to differences in the expression of the factors that are involved in Arpp21 degradation, for example the so far unknown E3 ubiquitin ligase. Future work will be required to understand these differences."

A key novel experiment is the modification of the ARPP21 locus to create a null allele and from there to assess the effect of loss of function on lymphocyte phenotype. This is a challenging task because the locus also harbors a microRNA (miR-128-2). The authors claim that there is no significant effect of their mutation on the expression of miR-128-2. This is based on the PCR analysis of whole thymus. However, while not statistically significant (NB I could not find any indication of what the statistical analysis was), miR-128-2 expression shows a strong decreasing trend upon Arpp21 KO. Because of the cell population sampled, it is not ruled out that miR-128-2 is more strongly affected in DN3 or DP thymocytes which express the greater amounts of host transcript.

Thank you for raising an important point. However, in thymocytes we observed a less than 2-fold reduction of miR-128, which can be judged as a modest change for miRNAs. Secondly, despite $n=7$ for both WT and KO thymocytes, the data were not significant (Student t-test, we have now provided this information in the Figure legend). As requested, we now show a detailed analysis of miR-128 expression across the thymic progenitors. Please note that mature miR-128-1 and miR-128-2 are identical in sequence so that miR-128 levels are always the sum of expression from the two miR-128 genes. These increased efforts clearly support our previous assertion that miR-128 levels are not significantly affected, as determined by two-way ANOVA. Data are depicted in **Extended Data Figure E3a-b**.

A previous study noted a substantial overlap between transcripts with binding sites for miR-128 and ARPP21, with common targets but opposing functions of ARPP21 and miR128. This complex interaction and uncertainty about whether the amounts of miR-128-2 in DN3 or DP cells might be different, and therefore influencing gene expression, needs further clarification. This is important for the mechanistic understanding of thymic development because whether an ARPP21/MiR182 axis is biologically relevant in thymus as in neuronal development (REF 14) is of interest and relevant to the current manuscript.

The previous analyses used different cell types and overexpression systems to determine Arpp21 targets. As requested, we have addressed the overlap of targets in thymocytes and have determined the presence of miR-128 binding sites in the mRNAs bound by Arpp21 as compared to their presence in mRNAs that were not bound by Arpp21. The enrichment of miRNA target genes in Arpp21 target genes by was calculated as follows: The thymus transcriptome was defined as genes detected in GSE242306 wildtype RNAseq samples (GSM7757939, GSM7757940, GSM7757941), miRNA target genes were obtained from the miRDB version 6.0 (<https://mirdb.org/download.html>) and Arpp21 targets as defined by this study. Classifying the thymocyte expressed genes as miRNA targets or non-targets as well as Arpp21-bound or unbound mRNAs we tested an association using fisher-test. In conclusion, we find a moderate enrichment of miR-128-3p targets in Arpp21-bound mRNAs (as reported before), however, this enrichment could also be observed for miR-181a-5p (see new **Extended data Fig. E3d**), as well as other miRNAs (not shown). This finding suggests that Arpp21 may to a certain extent promote the expression of potentially unstable mRNAs, since they tend to also be targets of miRNAs, but in the context of an unchanged miR-128 expression (see above and **Extended data Fig. E3a-b**) and the lack of conserved miR-128 binding sites in the Rag1 3'-UTR, it argues against a selective Arpp21/miR-128 axis of gene regulation in thymocytes. We have added this aspect into the results section: *"Bound mRNAs displayed a moderate enrichment for miR-128-3p binding sites, as previously suggested¹⁴, but also for the hematopoietic and thymus-expressed miRNA miR-181a^{25,26}, indicating that Arpp21 per se may preferentially recognize mRNAs subject to miRNA-dependent regulation (Extended Data Fig. 3d)." We thank the Reviewer for this comment and agree that addressing this point strengthens our conclusions.*

It is striking that the modification of the Arpp21 locus causes a strong increase of the short ARPP21 protein isoform (Fig 3B), which was barely detectable in the WT thymus. Although the short protein doesn't have RNA binding domains, it is possible that this protein contributes to the phenotype reported here and this is not addressed. Is the mRNA encoding the short isoform, like that of the long isoform, a target of miR-128-2?

The 3'-UTR of the short and long Arpp21 isoforms are encoded by different exons, they differ in sequence and the short isoform does not have a predicted miR-128 binding site:

https://www.ensembl.org/Mus_musculus/Transcript/Exons?db=core;g=ENSMUSG00000032503;r=9:111894159-112065006;t=ENSMUST00000178410

https://www.ensembl.org/Mus_musculus/Transcript/Exons?db=core;g=ENSMUSG00000032503;r=9:111894159-112065006;t=ENSMUST00000164754

Since the functional importance of the Arpp21^{short} isoform is currently unclear, we were unable to directly test a potential contribution. However, we can exclude an effect of Arpp21^{short} on the expression level of targets of Arpp21 (see **Extended data Fig. 3g**). Testing the physiologic contribution of Arpp21^{short} will involve the generation of yet another mouse model, which may be more relevant for neurobiology. However, such investigations go beyond the scope of this study, which focuses on the RNA-binding function of Arpp21 in thymocytes.

The NMR studies in Figure 4 and supplemental 5 reveal a domain-intrinsic specificity for uridine-rich sequences. This is consistent with the results of the CLIP in this manuscript and with a published study (cited by the authors as Ref-14) which concluded ARPP21 recognition of uridine-rich sequences with high specificity for 3'UTR. The ability of R3H domains to bind RNA is known and the experiments shown here confirm this is also the case for ARPP21.

We respectfully disagree with this assessment. We would argue that the R3h domain is known to have putative nucleic acid binding activity, interacting with RNA or DNA or mononucleotides (dGMP) or with the 5'-end of single-stranded DNA (Jaudzems et al., JMB 2012). The authors of Ref14 (Rehfeld et al., 2018) were the first to investigate the RNA-binding property of Arpp21. Using ectopic overexpression of truncated proteins in HEK 293 cells this study suggested that two domains, the R3H and the SUZ domains contact RNA (as assessed by UV-crosslinking of cells and radiolabeling of immunoprecipitated proteins/RNA adducts). From this finding a bipartite binding motif was expected, and the U-rich binding motif identified in CLIP could either relate to a preferential interaction of the R3H or SUZ domain or even involve another RBP that is present in the transfected cells. Our study is the first to investigate the intrinsic RNA-binding properties of recombinant Arpp21 protein. We show a direct interaction of the R3H domain of Arpp21 with RNA, we delineate the boundaries of the RNA-binding domain and rule out obvious contributions from regions outside (and of the SUZ domain), using both NMR and ITC. Most excitingly, we demonstrate a strong R3H domain-intrinsic preference for oligo-U sequences, and this finding now explains the motif found in our endogenous Arpp21 CLIP dataset (as well as the motif in the CLIP data of Ref. 14).

The manuscript reports the results of ARPP21 CLIP on thymocytes which identifies 6779 direct targets. The analysis intersects these with genes that are differentially expressed in an RNA-seq experiment comparing DN2/3 thymocytes from KO and WT. From here the focus shifts to the regulation of the Rag1 mRNA which encodes a protein essential for VDJ recombination.

The authors show that Rag1 mRNA contains a 3421 nucleotide long 3'UTR and map out the functional interactions using a reporter system which are extensive and consistent with the profile of RNA binding determined by CLIP. They then generate a mouse in which the majority of the Rag1 3'UTR is removed. This approach can be challenging to interpret as the interactions between the rag1 3'UTR with other factors are also lost. While the phenotype presented appears similar to the ARPP21 KO, more specific UTR mutations that only blocked only the ARPP21/rag1 UTR interaction would be stronger evidence of a causal relationship.

We would like to point out that the current gold standard for functional interactions of RBPs with 3'-UTR is still the reporter assay with overexpressed proteins and 3'-UTR-fragments artificially fused to reporter genes, which are expressed from several DNA plasmids (see **Figure 6c, d**). As one among few studies testing transacting factor/cis-element interactions by making another mouse model, we have tried to interfere with the binding of the transacting factor Arpp21 to a cis-element in the 3'-UTR of its target *Rag1*. In theory one would like to inactivate only one specific binding site and relate the observed phenotype to one RBP/RNA interaction. However, in our experimental work it was not possible to pinpoint a short motif in the 3'-UTR as being responsible for the observed regulation of *Rag1* mRNA by Arpp21. In fact, our iClip data identified many crosslinked U-stretches in the *Rag1* 3'-UTR, coinciding with very strong Arpp21 binding at many positions, and our reporter assays strongly suggest that Arpp21 acts via cumulative binding to extended U-rich regions, as underscored by the increasing effects of larger deletions that successively removed more of the U-rich stretches. To perform the suggested analysis, one would have to execute a CRISPR/Cas9 mutagenesis screen in the mouse germline comparing deletions of decreasing size and to then move on to combinations of smaller deletions. We believe that there is considerable functional redundancy in the interaction sites of Arpp21-sensitive mRNAs, so that in the end the larger the deletion, the greater the effect. We argue that the likelihood of this trivial result would not justify the technically impractical, labor-intensive and costly approach of *in vivo* site-directed mutagenesis. We hope the reviewers agree and that in this case "the perfect should not be the enemy of the good". Furthermore, we would argue that our mouse model did not claim to precisely define the binding site(s) in the large *Rag1* 3'-UTR for Arpp21 regulation *in vivo*. We rather asked whether rendering the 3'-UTR of *Rag1* insensitive to Arpp21 can phenocopy Arpp21 inactivation. We are very pleased to confirm this by the overlapping changes in thymocyte development, similar changes in TCR gene segment recombination and a similar reduction in Rag1 protein levels. We find it unlikely that this extent of phenocopy could be caused by inactivation of the binding site of another factor in the 3'-UTR of the *Rag1* mRNA that is similarly required to promote Rag1 expression but works independently of Arpp21. To satisfy the Reviewer's concerns, we have added this aspect into our results section: "*Taken together, these data connect the genetic inactivation of Arpp21 as well as the deletion of the Rag1 3'-UTR to reduced expression of Rag1 protein and show how this reduction is correlated with an increasing block in thymocyte development. The data suggest that the more Rag1 expression is reduced, the greater the inhibitory impact becomes on the transition of DN3 progenitors across β -selection.*"

The most novel aspect of this manuscript are the claims based around the thymic phenotype of the ARPP21 and Rag3' UTR mutant mice. The phenotype presented in Figures 5 A-C and 6A-D appears modest and more information is needed to clarify and strengthen the case for the interpretation the authors make. It will help to show cell numbers and to be clear about the characteristics of the mice being compared. The age and sex as well as whether littermates are being compared is relevant, it might also be helpful to have a quantitative analysis of cell numbers in heterozygous mice.

We agree that one prominent aspect among the many novel findings of this paper relates to the thymocyte phenotype.

To further support our notion that this is related to altered Rag1 expression and is most pronounced during the DN3a-DN3b transition, we performed a further single-cell analysis using high-resolution multi-parameter spectral flow cytometry. This allowed us to follow all stages of thymic development in one tube. Such a global overview of thymopoiesis confirmed our earlier notion. In our revised version, we have also added the cell numbers as requested. The mice analysed were littermates, mice of all genotypes were of comparable age. The new data are presented in the **new Figure 9** and **Extended data Figure 9**. We also believe that in the field of post-transcriptional gene regulation, the

magnitude of the observed phenotype is not the most relevant criterion. We hope that the value of our work is to characterize and quantify the contribution of a novel regulatory interaction to an essential step in T cell differentiation *in vivo*, and modestly suggest that our work is one of the few in the field to accomplish this degree of precision.

The results with early B cell development reports a small effect on the frequency of preB cells, but the reported results do not include the detection of cytoplasmic IgH which is a key relevant marker for these cells and this phenotype. This analysis would also benefit if cell numbers were presented.

We thank this reviewer for suggesting this new staining, which helped us to prove that Arpp21 does not significantly affect B cell development. As requested, we analysed the expression of intracellular IgM in early B cell precursors. In the bone marrow, in contrast to the thymus, we cannot detect significant Arpp21 protein expression (new **Extended data Figure E7d**) and Arpp21 deficiency does not result in a significant reduction of Rag1 mRNA (as already shown in **Extended data Figure 7e**), nor does it affect the expression of i.c. IgM (new **Extended data Figure 7f**). There is no significant difference in either bone marrow or thymic cellularity as shown in the new **Extended Data Figure E9a-b**.

The effects on Rag1 protein amounts are estimated by Western blot of whole thymus (Fig 6B and 8F) are variable between the ARPP21 and Rag3' UTR mouse models and there is some inconsistency between the cell types measured by in the different models. It is not clear if reduced amounts of Rag1 protein account for the phenotype reported. Given the reduced amounts in 6b are about half of the wildtype, one can predict Rag heterozygous mice should have a similar phenotype. Its not clear if that has ever been looked at.

We agree with this reviewer that it is difficult to estimate or agree on the actual amount of protein expression after visual inspection of Western blot signals. To address effects of heterozygosity, we have generated Rag1^{+/-} as well as Rag1^{3'del/-} mice to assess whether the reduced Rag1 expression resulting from gene dosage is comparable to that in Arpp21 KO mice (**Figure 9a** and **b**). To compare the Rag1 expression in these different mouse models, we have performed additional immunoblots with different amounts of WT protein extract and have included extracts from thymocytes of animals from all five genotypes on the same gel. Here we show that Rag1 protein expression in Rag1^{+/-} thymocytes is reduced to approximately half the amount visible in extracts from WT littermates. In Arpp21-deficient thymocytes, Rag1 expression is further reduced and does not exceed 25% of the amount found in extracts from WT littermates, and even lower expression was present in Rag1^{3'del/-} thymocytes. Analyzing thymocyte development in mice from each of these genotypes in parallel, we find similar block at the DN3a stage in Arpp21 and Rag mutants with the Rag1^{3'del/-} genotype being the most affected (**Figure 9** and **Extended data Fig. 9**). We hope that this additional evidence in support of our original conclusions satisfies the reviewer and thank him/her for a constructive suggestion.

It is unclear how a reduction in the amount of RAG1 leads to reduced clonotype diversity. Once a Rag induced break is made, the N additions etc that introduce junctional diversity shouldn't be impacted. Is it possible that ARPP21 has some impact on this process?

We apologise for the lack of clarity in the original version of our manuscript. We have made appropriate changes and added the following sentence "Given the limited time thymic progenitors have to successfully recombine TCR beta and alpha loci to escape apoptosis, combined with the sequential and highly orchestrated order of recombination element selection, altered Rag1 protein expression should result in a skewed recombination process and consequently an altered TCR repertoire."

Reduced levels of Rag1 in Arpp21 KO and Rag1 3'-del mice result in insufficient induction of DNA breaks during TCR recombination. Given the limited lifespan during which DN3 and DP thymocytes must successfully recombine their TCRB and TCRA loci to avoid apoptosis, this could have two consequences: First, as the recombination events are orchestrated according to V-D-J or V-J location on the respective chromosomes, the altered DNA break induction will "force" such thymocytes to use the most accessible fragments for recombination. In the TRAV locus, which is particularly long, the use of such readily accessible fragments is the most visible, hence the preferential use of the proximal V and distal J fragments, as observed in **Figure 7e, f** and **Figure 8h, i** and new **Extended Data Figure E8f, g**.

Second, if the reduction of Rag1 expression is severe enough, it should delay the recombination process, leading to the premature death of a higher proportion of progenitors in Arpp21- and Rag1 3'-UTR-deficient mice compared to WT counterparts. Since the thymus size of such KO mice is not reduced, KO thymocytes probably undergo some more vigorous proliferative expansion than WT counterparts do. Increased expansion of post-beta selection thymocytes should in turn lead to the reduced repertoire diversity that we have observed, particularly at the TCRB locus.

Should the effect of Arpp21-deficiency be more far-reaching than the reduction of Rag1 expression, we would expect an altered length of CDR3 fragments of TRBV and TRAV. However, this is not the case (**Fig. 7i** and **Extended Data Figure 8h**). These data are consistent with the observation made for human patients expressing Rag hypomorphs, where only repertoire complexity and V fragment usage were altered, but not CDR3 length [doi:10.1126/sciimmunol.aah6109].

In the Arpp21 deficient mice there is a phenotype at the beta selection checkpoint but not at the double positive stage in terms of cell numbers. The authors argue that premature downregulation of RAG1 leads to reduced recombination mainly of the beta chain. The relevance of the expression of Arpp21 in DP thymocytes remains uncertain and it is not clear why the reduced RAG1 expression in DP thymocytes does not lead to detectable changes in Valpha chain

rearrangement.

There is a qualitative difference for TRAV rearrangement, where the relevance of reduced Rag1 expression in DP thymocytes leads to the preferential use of proximal TRAV fragments by Arpp21 deficient thymocytes. At this point, we can only speculate that since recombination of the TCR alpha locus is a one-step process, and since both alleles are recombined simultaneously (thus doubling the chance of successful recombination), reduced Rag1 protein expression may manifest itself in a milder form - not as an overall failure, but as a skewed use of V and J fragments. This is all the more observable because the TRAV locus is long and the use of specific V and J fragments is well orchestrated and temporally separated. Similar observations are known from TCR sequencing of human Omenn syndrome patients: (Bosticardo, Pala and Notarangelo, Eur J. Immunol 2021, DOI: 10.1002/eji.202048880) "...Finally, a unique feature of V(D)J recombination at the TCRA locus is that it happens in waves, with initial rearrangements involving the most downstream TRAV and most upstream TRAJ genes. If such rearrangements are nonproductive, thymocytes proceed with further rearrangements that ultimately involve the most upstream TRAV and most downstream TRAJ genes. In patients with hypomorphic RAG mutations, there is reduced frequency of rearrangements involving these distal TCRA elements...".

The repertoire analysis for both mouse strains is performed on splenic CD4+ T cells. These data include 100- to 200-K sequences in each sample, giving a lot of statistical power, able, in principle, to detect small differences robustly. It is surprising that the effects of the Rag1 UTR mutation are smaller than those of Arpp21KO in terms of the effect on the clones (This is an indication the phenotype of the Arpp21 KO may result from additional effects). The split axis in 7b is confusing, it is not clear why it is included or needed, particularly as it's split differently for TCRVB on the left and TCRA on the right. I could not find, in the figure, or in the legend, whether this is a linear or a log scale. Nor was it clear what methodology was used to determine the values for the Shannon index in the graphs. There are two papers cited in the methods with no further explanation of which or how the tools were used. How does the analysis take into account the lower number of sequences in the Arpp21 ko?

There is indeed a high number of sequences detected, and we see this as an advantage, as illustrated by the TRAV diversity. Here, we can be more confident that there is no difference in sequence diversity despite the high number of sequences detected. For the same reason, we have split the axis (and shown it clearly) to better illustrate the difference in the TRBV repertoire and the lack of it for TRAV. We agree that the changes are rather modest (numerically), but similarly modest changes are seen in the human patients with hypomorphic Rag mutations and these changes result in an adaptive immune response that is sufficiently impaired to result in a readily detectable phenotype [i.e. some patients present with combined immune deficiency with granulomas or autoimmunity (CID-G/AI) and have a Shannon's index of their IGH and TRB, which are quite comparable to that of healthy donors [doi:10.1126/sciimmunol.aah6109]]. It may help to understand our argument by explaining the Shannon index better. This index is used in ecology to assess entropy and estimate species diversity. To describe populations, it takes into account both the number of species living in a habitat and their relative abundance. It has been adopted in immunology to assess TCR and BCR diversity, see for example, Lee et al. (doi:10.1126/sciimmunol.aah6109). For our use, the Shannon index takes into account the number of total sequences and the clone size distribution in the total repertoire. The scale of the index is logarithmic, we chose e as a base. The higher the number, the more diverse the repertoire. It was calculated with the following formula

$$H = -1 * \sum[(n/N) \times \ln(n/N)].$$

where

n - individuals of a given type (here TRAV or TRBV)

N - total number of individuals in a community (TRAV or TRBV).

A comprehensive overview of the method can be found at <https://www.omnicalculator.com/ecology/shannon-index>., Whenever the Shannon index is significantly altered for one chain of the TCR, we have to conclude that the repertoire diversity is altered.

The authors suggest that reduced Rag1 might decrease the efficiency of recombination and the VDJ/DJ ratio, (there is evidence for a reduced V-DJ with normal DJ on a gel in 6f), and could decrease the number of V(D)J clonotypes in a population, which does occur in the Arpp21 ko. It's unclear if the reduction in Rag1 can account for this

Reduced levels (concentration) of Rag1 in cells undergoing recombination leads to a lower frequency of induced DNA breaks. Since the recombination process is limited by time (progenitors have a defined window for successful recombination, (pre-)TCR assembly and selection, about 48 to 60 hours) (doi: 10.1016/j.it.2016.10.007.) and location of the recombined elements (certain V and J fragments are used earlier or later depending on their chromosomal location), altered Rag1 levels translate directly into a reduced repertoire. This phenomenon is well described for human Rag hypomorphs, where a mutation in either Rag loci leads to reduced Rag enzymatic activity (Villa A et al, 10.1016/s0092-8674(00)81448-8).

It is unclear why there is a difference between TCRB and TCRA and what this means. They mention that for TCRA, proximal V genes are used more when Rag expression is low, and their data agrees with this (7e, 7f). So that shifts the distribution of V genes used, but doesn't appear to impact on diversity, since in 7b, there is no difference in Shannon index for TCRA. The current presentation is thus rather unclear about TCR repertoire diversity per se and not convincing that this is more than reduced recombination. A role of Arpp21 in controlling positive

or negative selection in the thymus or additional function of the short isoform could be relevant and needs to be acknowledged when comparing these two mouse models as the phenotypes are clearly different.

We appreciate the reviewer's comment as it has prompted us to look at this issue more closely. The different output of TCRB and TCRA recombination when Rag1 is reduced may be a result of the organisation of these loci and the recombination process itself. TCRB is rearranged in two steps: D to J first, V to DJ later, whereas TCRA recombination is a single-step process. The TRVB locus is relatively short, about 700 kb, with most of the V fragments spread over a 235kb stretch of DNA (Wu G, doi/10.1073/pnas.2010077117 and <https://www.imgt.org/IMGTrepertoire/index.php?section=LocusGenes&repertoire=locus&species=mouse&group=TRB>). In contrast, the TRAV locus is much longer, spanning 1800 kb (<https://www.imgt.org/IMGTrepertoire/index.php?section=LocusGenes&repertoire=locus&species=mouse&group=TRA>). Thus, altered recombination of the TCRB locus tends to manifest itself in a reduced number of productive sequences, without much effect on which V fragment is used. The TCRA locus, on the other hand, is more affected by the use of the V fragment, as the V and J fragments are used in a highly coordinated manner, with a clear boundary between primary and secondary recombination events (Hawwari A et al, doi: 10.1038/ni1189, Abarrategui I & Krangel MS, doi: 10.1038/ni1379, Guo I et al, DOI: 10.1038/ni791).

We would like to point out that the overall concept of altered repertoire diversity in *Arpp21*- and Rag1 3'-UTR deficient mice is a sum of events occurring at the TCRB and TCRA loci. These recombination events are of course separated in time, but the quality of the TCRs generated depends on both. Therefore, we believe, and here we respectfully disagree with the reviewer, that these mutants need to be considered not only in the context of their Rag1 regulation, but in a broader perspective of how such reduced Rag1 expression alters TCR diversity and quality.

Overall, this is of interest because it appears to be a new mechanism that regulates Rag1, but the title is misleading, overstating the effects on TCR diversity – and suggesting that these are due to reduced Rag expression. Reduced diversity could also be due to reduction in one of the many other *Arpp21* targets, or other changes as the RAG1 UTR models shows a less strong phenotype.

Please note that *Arpp21* mainly enhances the translation of target mRNAs and this may potentially involve binding of *Arpp21* to additional sites on the *Rag1* mRNA, which we could not delete. For example, we did not want to affect mRNA integrity by deleting the most 5' and most 3' parts of this 3'-UTR, which could affect translation or polyadenylation. These aspects could explain somewhat milder phenotypes in this mutant when comparing both mouse models.

There are rare examples where one post-transcriptional regulator-target pair is functionally dominant, as illustrated in the work of Lu et al. (10.1084/jem.20140338) on the miR-155-PU.1 axis, explaining a phenocopy upon deletion of cis-element or transacting factor. In our work, the similarities between *Arpp21* KO and Rag 3'-UTR KO are striking (although somewhat milder in the latter) and illustrated in almost every aspect tested, i.e. progenitor frequency (**Figure 5d** and **Figure 8b, c**); TCR beta i.c. expression (**Figure 6e** and **Figure 8d**); oligoclonal expansion (**Figure 7d** and **Figure 8g**); Rag1 protein expression (**Figure 6b** and **8f** and **new Figure 9a**) as well as TRAV and TRAJ usage (**Figure 7h** and **Figure 8h,i, Extended data Figure 8f, g**).

We respectfully disagree to the assessment of the TCR repertoire diversity reduction and the representation of our data in the title. Deleting the main *Arpp21*-regulated region in the 3'-UTR of *Rag1* is state-of-the-art, and has not been done very often in our field of research. It should be noted that we generated a second mouse model and did indeed find a highly overlapping phenotype as listed above, which to our understanding is the best possible evidence for a causal relationship. Since altered Rag1 expression is sufficient to explain all of the observed phenomena we provide an explanation with the smallest possible set of elements. Certainly, we cannot exclude contributions from additional *Arpp21* targets, however, in sum we believe we have made a convincing case for Rag1 as the principal target for the phenotypes we describe.

There are some interesting parallels as the authors discuss with the Omenn syndrome and potential oligoclonality and the usage of more proximal TRAVa segments due to lower functionality of Rag. However, the two mouse models introduced here show no evidence of any T cell dysfunction, self-reactivity and the mice are healthy. Has the T cell biology been investigated? Is there a loss of T cell function/dysfunction? If none of this is the case then these mice might not be a good model for Omenn syndrome. We agree that the parallels of Omenn syndrome and our mouse model are worth mentioning in the discussion. Since we do not have direct proof for altered immune function in *Arpp21*-deficient mice, we have removed the half-sentence that the *Arpp21* ko could be a model for Omenn syndrome.

Minor comments

Line 367 indicates that the Rag13'-UTR-KO mice were generated by crossing of two founders to generate homozygous F1 offspring. Have unanticipated mutations in the rag1 locus introduced by cas9 been ruled out?

We have now had the chance to cross the founders to C57BL/6 mice to decrease the chances of off-target mutations. Together, we analyzed mice over several generations with and without backcrossing and these mouse lines maintained the observed phenotypes (**see new Figure 9**). The correct boundaries of 3'-UTR deletion mutations have been determined by Sanger sequencing.

Improve figure legends and methods. Particularly, include keys for colours and symbols in graphs and include the number of biological replicates when missing. Authors only show one representative western blot picture for each experiment; they could include a graph showing relative protein levels for all the biological replicates done in an experiment.

We improved the figure legends and methods and Western blot representations.

Lines 136-138 and Fig 1C: where do the 490 proteins come from?

490 proteins are RBPs which are confirmed in two ways (published RBP or cross-link in peptide sequence identified plus peptide in OOPS).

Fig 2C/2D: western blot pictures are cropped to close to the Arpp21 band, and it is not possible to see if there is the phosphorylated band in these blots.

The typical stimulation time of 2 hrs does not allow to see phosphorylated Arpp21 protein, unless MG132 is used (uncropped blots will of course be provided).

Fig 2F: there are no bands at 100 KDa in the ubiquitin blot following Arpp21 IP. There should be a band at the height of the protein immunoprecipitated. There is a faint signal just below 135 KDa. It would help if the authors point to the bands of interest with arrows.

We do not think that the unmodified Arpp21 protein should be detected by ubiquitin-specific antibodies and based on WB levels of mono- and di-ubiquitinated species are below the detection level in thymocytes.

Figure 4 e misses the 1 in Arpp21

Thank you for catching this- we have corrected the mistake.

Suppl. Fig 2b: there is an accumulation of the phosphorylated Arpp21 protein also upon Iono treatment which was never seen before; in fact, the phosphorylated band appeared upon Iono treatment and proteasome inhibition. We have mostly analyzed phosphorylation/ degradation at late time points (2h). However, in combination with the inhibitors, we went earlier (30min), because we wanted to minimize toxicity in combination with ionomycin treatment. At this time point, we can readily see P-Arpp21 (please also see our kinetics of ionomycin stimulation and Arpp21 phosphorylation degradation below).

Figure 1. Short-term stimulation enables the detection of phosphorylated Arpp21 protein. Thymocytes from WT and Arpp21-deficient mice were stimulated for the indicated time points with ionomycin. Protein extracts were analyzed by Western blot procedure.

We have now added this information to the methods section: "When thymocytes were pretreated with BTP2 or KN93 for 90 min, the stimulation with ionomycin was shortened to 30 min."

Correct Spelling in supplemental 6b

Thank you for catching this mistake, we corrected it.

Reviewer #2 (Remarks to the Author):

The noteworthy results of this manuscript are the discoveries that the RNA binding protein Arpp21 associates with the 3'UTR of Rag1, promoting its stabilization and role in TCR rearrangement. Loss of Arpp21, or removing the binding region in Rag1 3'UTR, results in a less diverse TCR repertoire. The methodologies employed are sound. On the whole, major claims (particularly those made early in the manuscript) are well-supported by the data. However, some minor weaknesses and points of confusion exist and should be addressed:

1. (Figure 5b and c) Authors claim that while overall numbers of developing T cells in the thymus are unchanged, there are reductions of cells at DN3 stage interpreted as a developmental block during beta chain rearrangement. a. Given that there is also an increase of cells at DN4 stage in the KO, it may also be interpreted that cells move through the DN3 stage more quickly and that the block may be later at the DN4 stage when pre-TCRb signal is required to progress. Is there evidence that more T cells are dying for lack of a functional pre-TCR signal? b. The differences in frequency are quite subtle and it may help to further interrogate the phenotype or function of thymic emigrants to demonstrate the biological significance of Arpp21 loss on T cell development.

We have improved the studies on thymocyte development by including not only frequencies but also absolute numbers of developing progenitors.

a. We would like to point out that it is the opposite - in Arpp21 or Rag1 3'-UTR deficient mice there is accumulation of cells at the DN3a stage (**Figure 5d** and **Figure 8b,c**) and reduced frequency of some progenitors after beta selection (**Figure 5d** and **Figure 8c**). The accumulation of progenitors just before beta selection led us to hypothesize that this was due to altered TCR beta recombination. As a crude test to verify this, we looked at the intracellular expression of TCRbeta (a surrogate measure of preTCR) and found an approximately 50% reduction in its expression in Arpp21- or Rag1 3'-UTR-deficient mice (**Figures 6e** and **Figure 8d**). Given that the survival of T cell progenitors at this stage depends on functional TCR signalling, we focused more on alterations in TCR recombination.

b. We agree that gating on recent thymic emigrants could be one way to come up with stronger phenotypes. Unfortunately, we were not able to cross in the RAG2-GFP mouse within a reasonable time frame. Importing this mouse from the US, crossing the lines and generating homozygous offspring would have taken much more than half a year to generate the new line.

2. (Figures 7 and 8) It is interesting that there are differences between TRBV and TRAV gene rearrangement in absence of Arpp21, namely, TRBV diversity is reduced while TRAV diversity is unchanged, and TRBV gene usage is unchanged while TRAV gene usage is skewed.

a. Are these observations related to the fact that TRB alleles are sequentially rearranged, while TRA alleles rearrange simultaneously (and are sometimes co-expressed)?

b. Diversity was not impacted by disrupting Rag1 3'UTR interaction with Arpp21 but gene usage skewing still exists. Might this suggest that the observed impact of Arpp21 loss on diversity or gene usage skewing is Rag1-independent? We are grateful to the reviewer for raising this point, as it prompted us to look more closely at the TRAV locus recombination. We have now incorporated the results of our extended investigation into the **Extended Data Figure E8f-h**.

a. Our interpretation of the different phenotypic outcome in TRBV and TRAV recombination is not due to allelic exclusion at the TRBV locus and lack of it in TRAV, but rather to the length of the loci and time given to complete the task. The TRAV locus is exceptionally long, spanning 1800 kb, and the selection of particular variable and joining fragments is highly coordinated. As the reviewer has already pointed out, both TRAV loci are recombined simultaneously, but the process occurs at the same rate (Hawwari A et al, doi: 10.1038/ni1189, Abarrategui I & Krangel MS, doi: 10.1038/ni1379, Guo I et al, DOI: 10.1038/ni791). Since the recombination of the TRB locus occurs in two steps, and the final V-DJ recombination occurs from a relatively short stretch of DNA (about 700 kb, with most of the V fragments spread over only 235 kb) the specific location of the V fragments play apparently a lesser role.

b. However, given the sequential order of events in the recombination of the TRAV locus (now extended to the use of TRAJ fragments) and the fact that the most affected fragments are those most distal to the TEA promoter, the phenomenon can be explained solely by a reduced level of Rag1 expression. Overall, the phenotype of the Rag1 3'-UTR deficiency is only somewhat milder than that of Arpp21, with a trend but no significant differences in Shannon's index (please note, however, that some oligoclonality can also be observed in the Rag1 3'-UTR KO, (**Figure 8g**)). To further demonstrate that the observed changes are the likely result of reduced Rag1 expression, we have combined the Rag13'-del allele with heterozygosity of the Rag1 knockout. Here we find a further reduction in Rag1 expression and the strongest block in thymocyte development at the DN3b stage (**Figure 9a, b and f, g**).

Both, the reduced diversity of productive TCR β rearrangement as well as gene segment skewing for the TCR α rearrangements were also observed in high-throughput TCR sequencing approaches on peripheral T cells from human patients with RAG mutations (Bosticardo, Pala and Notarangelo, Eur J. Immunol 2021, DOI: 10.1002/eji.202048880; Rowe et al., J Allergy Clin Immunol 2018; doi: 10.1016/j.jaci.2017.08.001).

Overall, the data that Arpp21 and Rag1 interact appear strong, but the functional consequences of this still remain a bit of a mystery.

We concur that the Arpp21-Rag1 axis is strongly supported and that the main features of the observed phenotypes can be explained by the stabilisation of Rag1 mRNA by Arpp21. As the Rag1 3'-UTR knockout phenotype on the TCR repertoire was somewhat weaker than that of Arpp21, it is possible that some additional Arpp21 function play a lesser role here as well. We hope that the inclusion of thymocyte development data from additional mouse lines with the graded reduction of Rag1 expression coupled to the UMAP analysis strengthen and clarify our description of the functional consequences of the Rag1-Arpp21 interaction.

Reviewer #3 (Remarks to the Author):

RNA binding proteins play complex and essential roles in gene regulation, but their functions are poorly understood. Xu and colleagues reveal the function of a known binding protein - Arpp21, which they show plays an important and previously unappreciated role in regulation of the Rag1 recombinase that rearranges T cell receptor genes during development in the thymus. The authors construct elegant Arpp21 deficient mice that leave the short isoform of Arpp21 intact, which is important as Arpp21short is important in neural development. These mice are shown to have a defect in TCR beta gene rearrangement and also in a process known as sequential TCR alpha gene rearrangement, which requires high persistent levels of RAG activity and is important to maximize the breadth of the TCR repertoire in T cells. The authors map the binding sites of Arpp21 in Rag1. They make a further elegant mouse mutant in the 3' site of RAG1 that binds Arpp21, and establish that this RAG1 3' UTR mutant phenocopies the Arpp21 mutant they characterized in its effects on RAG1 expression.

The work here represents an exceptionally thorough set of experiments. The experiments are well performed, the writing is clear and concise. One minor comment is that a similar scheme of Arpp21 and Arpp21short added to Figure 1 would help the reader. Outside this detail which the authors are free to ignore, I really have no substantive criticisms or suggestions for the authors. I wish to congratulate them on their interesting, thorough and well-performed study.

[redacted]

[editorial note: reviewer name redacted]

We have included the scheme in Fig. 1e.

We are very thankful for the enthusiasm this reviewer expresses about our work and hope that this appraisal is given appropriate weight during the editorial decision.

REVIEWERS' COMMENTS

Reviewer #1 (Remarks to the Author):

The revised manuscript extends the insightful work of the original version highlighting the new finding in Rag1 heterozygosity which shows an effect when RAG amounts are reduced by approximately twofold. This is a new interesting and supportive finding. The clarification of the dichotomy between T and B cells for their requirement for ARPP21 is also of interest.

I am very supportive of the approach to engineering noncoding regions used here, which is not widely pursued in vivo. These are complex experiments to perform and interpret. The formal possibility that ARPP21 may have other effects that contribute to the phenotype is acknowledged, but it is now clearer from the revised manuscript that this is unlikely. Reversal of normal phenotype upon restoration of Rag1 levels would be a gold standard, but this is exceedingly difficult to achieve with precision in this system.

Minor points

Confirm the intracellular IgM referred to in ED 7F is in fact Ig mu-heavy chain.

Reviewer #2 (Remarks to the Author):

Thanks to the authors for their thorough consideration of, and detailed responses to, the points I raised. I look forward to the future functional studies for which your work provides rationale.

Reviewer #3 (Remarks to the Author):

I continue to be strongly supportive of this interesting and well-performed work.

Point-by-point response:

Reviewer #1 (Remarks to the Author):

The revised manuscript extends the insightful work of the original version highlighting the new finding in Rag1 heterozygosity which shows an effect when RAG amounts are reduced by approximately twofold. This is a new interesting and supportive finding. The clarification of the dichotomy between T and B cells for their requirement for ARPP21 is also of interest.

I am very supportive of the approach to engineering noncoding regions used here, which is not widely pursued in vivo. These are complex experiments to perform and interpret. The formal possibility that ARPP21 may have other effects that contribute to the phenotype is acknowledged, but it is now clearer from the revised manuscript that this is unlikely. Reversal of normal phenotype upon restoration of Rag1 levels would be a gold standard, but this is exceedingly difficult to achieve with precision in this system.

Minor points

Confirm the intracellular IgM referred to in ED 7F is in fact Ig mu-heavy chain.

Yes, indeed we confirm, we used Fluorescein (FITC) AffiniPure™ Fab Fragment Goat Anti-Mouse IgM, μ chain specific from Jackson ImmunoResearch. The company states: "Based on antigen-binding assay and/or ELISA, the antibody reacts with the heavy chain of mouse IgM but not with mouse IgG or the light chains of mouse immunoglobulins. No antibody was detected against non-immunoglobulin serum proteins. The antibody may cross-react with IgM from other species."

Reviewer #2 (Remarks to the Author):

Thanks to the authors for their thorough consideration of, and detailed responses to, the points I raised. I look forward to the future functional studies for which your work provides rationale.

Reviewer #3 (Remarks to the Author):

I continue to be strongly supportive of this interesting and well-performed work.